# Enhancing the Hierarchical Environment Design via Generative Trajectory Modeling

## Abstract

Unsupervised Environment Design (UED) is a paradigm that automatically generates a curriculum of training environments, enabling agents trained in these environments to develop general capabilities, i.e., achieving good zero-shot transfer performance. However, existing UED approaches focus primarily on the random generation of environments for open-ended agent training. This is impractical in resource-limited scenarios where there is a constraint on the number of environments that can be generated. In this paper, we introduce a hierarchical MDP framework for environment design under resource constraints. It consists of an upper-level RL teacher agent that generates suitable training environments for a lower-level student agent. The RL teacher can leverage previously discovered environment structures and generate environments at the frontier of the student's capabilities by observing the student policy's representation. Additionally, to alleviate the time-consuming process of collecting the experience of the upper-level teacher, we utilize recent advances in generative modeling to synthesize a trajectory dataset for training the teacher agent. Our method significantly reduces the resource-intensive interactions between agents and environments, and empirical experiments across various domains demonstrate the effectiveness of our approach.

## 1 Introduction

The advances of reinforcement learning (RL) [17] have promoted research into the problem of training autonomous agents that are capable of accomplishing complex tasks. One interesting, yet underexplored, area is training agents to perform well in unseen environments, a concept referred to as zero-shot transfer performance. To this end, Unsupervised Environment Design (UED) [3] has emerged as a promising paradigm to address this problem. The objective of UED is to automatically generate environments in a curriculum-based manner, and training agents in these sequentially generated environments can equip agents with general capabilities, enabling agents to learn robust and adaptive behaviors that can be transferred to new scenarios without explicit exposure during training.

Existing approaches in UED primarily focus on building an adaptive curriculum for the environment generation process to train the generally capable agent. Dennis et al. [3] formalize the problem of finding adaptive curricula through a game involving an adversarial environment generator (teacher agent), an antagonist agent (expert agent), and the protagonist agent (student agent). The RL-based teacher is designed to generate environments that maximize regret, defined as the difference between the protagonist and antagonist agent's expected rewards. They show that these agents will reach a Nash Equilibrium where the student agent learns the minimax regret policy. However, since the teacher agent adapts solely based on the regret feedback, it is inherently difficult to adapt to student

policy changes. Meanwhile, training such an RL-based teacher remains a challenge because of the high computational cost of training an expert antagonist agent for each environment.

In contrast, domain randomization [19] based approaches circumvent the overhead of developing an RL teacher by training agents in randomly generated environments, resulting in good empirical performances. Building upon this, Jiang et al. [7] introduce an emergent curriculum by sampling randomly generated environments with high regret value [1] to train the agent. Parker-Holder et al. [10] then propose the adaptive curricula by manually designing a principled, regret-based curriculum, which involves generating random environments with increasing complexity. While these domain randomization-based algorithms have demonstrated good zero-shot transfer performance, they face limitations in efficiently exploring large environment design spaces and exploiting the inherent structure of previously discovered environments. Moreover, existing UED approaches typically rely on open-ended learning, necessitating a long training horizon, which is unrealistic in the real world due to resource constraints. Our goal is to develop a teacher policy capable of generating environments that are perfectly matched to the current skill levels of student agents, thereby allowing students to achieve optimal general capability within a strict budget for the number of environments generated and within a shorter training time horizon.

In this paper, we address these challenges by introducing a novel, adaptive environment design framework. The core idea involves using a hierarchical Markov Decision Process (MDP) to simultaneously formulate the evolution of an upper-level teacher agent, tasked with generating suitable environments to train the lower-level student agent to achieve general capabilities. To accurately guide the generation of environments at the frontier of the student agent's current capabilities, we propose approximating the student agent's policy/capability by its performances across a set of diverse evaluation environments, which acts as the state abstraction for the teacher's decision-making process. The transitions in the teacher's state represent the trajectories of the student agent's capability after training in the generated environment. However, collecting experience for the upper-level teacher agent is slow and resource-intensive, since each upper-level MDP transition evolves a complete training cycle of the student agent on the generated environment. To accelerate the collection of upper-level MDP experiences, we utilize advances in diffusion models that can generate new data points capturing complex distribution properties, such as skewness and multi-modality, exhibited in the collected dataset [11]. Specifically, we employ diffusion probabilistic model [15, 6] to learn the evolution trajectory of student policy/capability and generate synthetic experiences to enhance the training efficiency of the teacher agent. Our method, called *Synthetically-enhanced Hierarchical Environment Design* (*SHED*), automatically generates increasingly complex environments suited to the current capabilities of student agents.

In summary, we make the following contributions:

- We develop a novel hierarchical MDP framework for UED that introduces a straightforward method to represent the current capability level of the student agent.

- We introduce *SHED*, which utilizes diffusion-based techniques to generate synthetic experiences. This method can accelerate the training of the off-policy teacher agent.

- We demonstrate that our method outperforms existing UED approaches (i.e., achieving a better general capability under resource constraints) in different task domains.

## 2 Preliminaries

In this section, we provide an overview of two main research areas upon which our work is based.

### 2.1 Unsupervised Environment Design

The objective of UED is to generate a sequence of environments that effectively train the student agent to achieve a general capability. Dennis et al. [3] first model UED with an Underspecified Partially Observable Markov Decision Process (UPOMDP), which is a tuple

$$\mathcal{M} = \; < A, O, \Theta, S^{\mathcal{M}}, \mathcal{P}^{\mathcal{M}}, \mathcal{I}^{\mathcal{M}}, \mathcal{R}^{\mathcal{M}}, \gamma >$$

---

[1]They approximate the regret value by the Generalized Advantage Estimate [12].

. The UPOMDP has a set $\Theta$ representing the free parameters of the environments, which are determined by the teacher agent and can be distinct to generate the next new environment. Further, these parameters are incorporated into the environment-dependent transition function $\mathcal{P}^{\mathcal{M}} : S \times A \times \Theta \to S$. Here $A$ represents the set of actions, $S$ is the set of states. Similarly, $\mathcal{I}^{\mathcal{M}} : S \to O$ is the environment-dependent observation function, $\mathcal{R}^{\mathcal{M}}$ is the reward function, and $\gamma$ is the discount factor. Specifically, given the environment parameters $\vec{\theta} \in \Theta$, we denote the corresponding environment instance as $\mathcal{M}_{\vec{\theta}}$. The student policy $\pi$ is trained to maximize the cumulative rewards $V^{\mathcal{M}_{\vec{\theta}}}(\pi) = \sum_{t=0}^{T} \gamma^t r_t$ in the given environment $\mathcal{M}_{\vec{\theta}}$ under a time horizon $T$, and $r_t$ are the collected rewards in $\mathcal{M}_{\vec{\theta}}$. Existing works on UED consist of two main strands: the RL-based environment generation approach and the domain randomization-based environment generation approach.

The RL-based generation approach was first formalized by Dennis et al. [3] as a self-supervised RL paradigm for generating environments. This approach involves co-evolving an environment generator policy (teacher) with an agent policy $\pi$ (student), where the teacher's role is to generate environment instances that best support the student agent's continual learning. The teacher is trained to produce challenging yet solvable environments that maximize the regret measure, which is defined as the performance difference between the current student agent and a well-trained expert agent $\pi^*$ within the current environment: $Regret^{\mathcal{M}_{\vec{\theta}}}(\pi, \pi^*) = V^{\mathcal{M}_{\vec{\theta}}}(\pi^*) - V^{\mathcal{M}_{\vec{\theta}}}(\pi)$.

The domain randomization-based generation approach, on the other hand, involves randomly generating environments. Jiang et al. [7] propose to collect encountered environments with high learning potentials, which are approximated by the Generalized Advantage Estimation (GAE) [12], and then the student agent can selectively train in these environments, resulting in an emergent curriculum of increasing difficulty. Additionally, Parker-Holder et al. [10] adopt a different strategy by using predetermined starting points for the environment generation process and gradually increasing complexity. They manually divide the environment design space into different difficulty levels and employ human-defined edits to generate similar environments with high learning potentials. Their algorithm, ACCEL, is currently the state-of-the-art (SOTA) in the field, and we use an edited version of ACCEL as a baseline in our experiments.

## 2.2 Diffusion Probabilistic Models

Diffusion models [15] are a specific type of generative model that learns the data distribution. Recent advances in diffusion-based models, including Langevin dynamics and score-based generative models, have shown promising results in various applications, such as time series forecasting [18], robust learning [9], anomaly detection [21] as well as synthesizing high-quality images from text descriptions [8, 11]. These models can be trained using standard optimization techniques, such as stochastic gradient descent, making them highly scalable and easy to implement.

In a diffusion probabilistic model, we assume a $d$-dimensional random variable $x_0 \in \mathbb{R}^d$ with an unknown distribution $q(x_0)$. Diffusion Probabilistic model involves two Markov chains: a predefined forward chain $q(x_k|x_{k-1})$ that perturbs data to noise, and a trainable reverse chain $p_\phi(x_{k-1}|x_k)$ that converts noise back to data. The forward chain is typically designed to transform any data distribution into a simple prior distribution (e.g., standard Gaussian) by considering perturb data with Gaussian noise of zero mean and a fixed variance schedule $\{\beta_k\}_{k=1}^K$ for $K$ steps:

$$q(x_k|x_{k-1}) = \mathcal{N}(x_k; \sqrt{1-\beta_k}x_{k-1}, \beta_t \mathbf{I}) \quad \text{and} \quad q(x_{1:K}|x_0) = \Pi_{k=1}^K q(x_k|x_{k-1}), \quad (1)$$

where $k \in \{1, \ldots, K\}$, and $0 < \beta_{1:K} < 1$ denote the noise scale scheduling. As $K \to \infty$, $x_K$ will converge to isometric Gaussian noise: $x_K \to \mathcal{N}(0, \mathbf{I})$. According to the rule of the sum of normally distributed random variables, the choice of Gaussian noise provides a closed-form solution to generate arbitrary time-step $x_k$ through:

$$x_k = \sqrt{\bar{\alpha}_k}x_0 + \sqrt{1 - \bar{\alpha}_k}\epsilon, \quad \text{where} \quad \epsilon \sim \mathcal{N}(0, \mathbf{I}). \quad (2)$$

Here $\alpha_k = 1 - \beta_k$ and $\bar{\alpha}_k = \prod_{s=1}^k \alpha_s$. The reverse chain $p_\phi(x_{k-1}|x_k)$ reverses the forward process by learning transition kernels parameterized by deep neural networks. Specifically, considering the Markov chain parameterized by $\phi$, denoising arbitrary Gaussian noise into clean data samples can be written as:

$$p_\phi(x_{k-1}|x_k) = \mathcal{N}(x_{k-1}; \mu_\phi(x_k, k), \Sigma_\phi(x_k, k)) \quad (3)$$

It uses the Gaussian form $p_\phi(x_{k-1}|x_k)$ because the reverse process has the identical function form as the forward process when $\beta_t$ is small [15]. Ho et al. [6] consider the following parameterization of

$p_\phi(x_{k-1}|x_k)$:

$$\mu_\phi(x_k, k) = \frac{1}{\alpha_k}\left(x_k - \frac{\beta_k}{\sqrt{1-\alpha_k}}\epsilon_\phi(x_k, k)\right) \text{ and } \Sigma_\phi(x_k, k) = \tilde{\beta}_k^{1/2} \text{ where } \tilde{\beta}_k = \begin{cases} \frac{1-\alpha_{k-1}}{1-\alpha_k}\beta_k & k > 1 \\ \beta_1 & k = 1 \end{cases}$$
(4)

$\epsilon_\phi$ is a trainable function to predict the noise vector $\epsilon$ from $x_k$. Ho et al. [6] show that training the reverse chain to maximize the log-likelihood $\int q(x_0)\log p_\phi(x_0)dx_0$ is equivalent to minimizing re-weighted evidence lower bound (ELBO) that fits the noise. They derive the final simplified optimization objective:

$$\mathcal{L}(\phi) = \mathbb{E}_{x_0, k, \epsilon}\left[\|\epsilon - \epsilon_\phi(\sqrt{\bar{\alpha}_k}x_0 + \sqrt{1-\bar{\alpha}_k}\epsilon, k)\|^2\right].$$
(5)

Once the model is trained, new data points can be subsequently generated by first sampling a random vector from the prior distribution, followed by ancestral sampling through the reverse Markov chain in Equation 3.

## 3   Approach

In this section, we formally describe our method, *S*ynthetically-enhanced *H*ierarchical *E*nvironment *D*esign (*SHED*), which is a novel framework for UED under resource constraints. The *SHED* incorporates two key components that differentiate it from existing UED approaches:

- A hierarchical MDP framework to generate suitable environments,

- A generative model to generate the synthetic trajectories.

*SHED* uses a hierarchical MDP framework where an RL teacher leverages the observed student's policy representation to generate environments at the student's capabilities frontier. Such targeted environment generation process enhances the student's general capability by utilizing the underlying structure of previously discovered environments, rather than relying on the open-ended random generation. Besides, *SHED* leverages advances in generative models to generate synthetic trajectories that can be used to train the off-policy teacher agent, which significantly reduces the costly interactions between the agents and the environments. The overall framework is shown in Figure 1, and the pseudo-code is provided in Algorithm 1.

### 3.1   Hierarchical Environment Design

The objective is to generate a limited number of environments that are designed to enhance the general capability of the student agent. Inspired by the principles of PAIRED [3], we adopt an RL-based approach for the environment generation process. To better generate suitable environments tailored to the current student skill level, *SHED* uses the hierarchical MDP framework, consisting of an upper-level RL teacher policy $\Lambda$ and a lower-level student policy $\pi$. Specifically, the teacher policy, $\Lambda : \Pi \to \Theta$, maps from the space of all potential student policies $\Pi$ to the space of environment parameters $\Theta$. Existing RL-based methods (e.g., PARIED) rely solely on regret feedback and fail to effectively capture the nuances of the student policy. To address this challenge, *SHED* enhances understanding by encoding the student policy $\pi$ into a vector that serves as the state abstraction for teacher $\Lambda$. Rather than compressing the knowledge in the student policy network, we approximate the embedding of the student policy $\pi$ by assessing performance across a set of diverse evaluation environments. This performance vector, denoted as $p(\pi)$, gives us a practical estimate of the student's current general capabilities, enabling the teacher to customize the next training environments accordingly. In our hierarchical framework, the environment generation process is governed by discrete-time dynamics. We delve into the specifics below.

**Upper-level teacher MDP**. The upper-level teacher operates at a coarser layer of student policy abstraction and generates environments to train the lower-level student agent. This process can be formally modeled as an MDP by the tuple $< S^u, A^u, P^u, R^u, \gamma^u >$:

- $S^u$ represents the upper-level state space. Typically, $s^u = p(\pi) = [p_1, \ldots, p_m]$ denotes the student performance vector across $m$ diverse evaluation environments. This vector serves as the representation of the student policy $\pi$ and is observed by the teacher.

**Algorithm 1** *SHED*

---

**Input:** real data ratio $\psi \in [0, 1]$, evaluate environment set $\theta^{\text{eval}}$, reward function $R$;

1: **Initialize:** diffusion model $D$, teacher policy $\Lambda$, real and synthetic replay buffer $\mathcal{B}_{\text{real}}, \mathcal{B}_{\text{syn}} = \emptyset$;

2: **for** episode $ep = 1, \ldots, K$ **do**

3:     Initialize student policy $\pi$

4:     Evaluate $\pi$ on $\theta^{\text{eval}}$ and get state $s^u = p(\pi)$

5:     **for** Budget $t = 1, \ldots, T$ **do**

6:         generate $\vec{\theta} \sim \Lambda$, and create $\mathcal{M}_{\vec{\theta}}(\pi)$

7:         train $\pi$ on $\mathcal{M}_{\vec{\theta}}$ to maximize $V^{\vec{\theta}}(\pi)$

8:         evaluate $\pi$ on $\theta^{\text{eval}}$ and get next state $s'$

9:         compute teacher's reward $r_t$ according to $R$

10:        add experience $(s_t^u, \vec{\theta}, r_t^u, s_t^{u,\prime})$ to $\mathcal{B}_{real}$

11:        train $D$ with samples from $\mathcal{B}_{\text{real}}$

12:        generate synthetic experiences from $D$ and add them to $\mathcal{B}_{\text{syn}}$

13:        train $\Lambda$ on samples from $\mathcal{B}_{\text{real}} \bigcup \mathcal{B}_{\text{syn}}$ mixed with ratio $\psi$

14:        set $s = s'$;

15:     **end for**

16: **end for**

**Output:** $\Lambda, \pi, D$

---

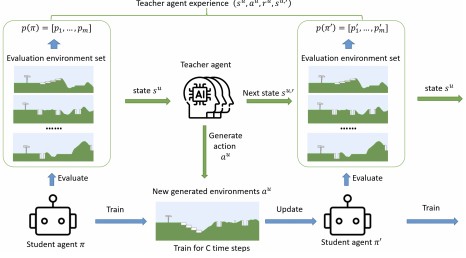

Figure 1: The overall framework of *SHED*.

Figure 2: The illustration of the environment generation process.

- $A^u$ is the upper-level action space. The teacher observes the abstraction of the student policy, $s^u$ and produces an upper-level action $a^u$ which is the environment parameters $\vec{\theta}$. $\vec{\theta}(a^u)$ is then used to generate specific environment instances $\mathcal{M}_{\vec{\theta}}$. Thus the upper-level action space $A^u$ is the environment parameter space $\Theta$.

- $P^u$ denotes the action-dependent transition dynamics of the upper-level state. The general capability of the student policy evolves due to training the student agent on the generated environments.

- $R^u$ provides the upper-level reward to the teacher at the end of training the student on the generated environment. The design of $R^u$ will be discussed in Section 3.3.

As shown in Figure 2, given the student policy $\pi$, the teacher $\Lambda$ first observes the representation of the student policy, $s^u = [p_1, \ldots, p_m]$. Then teacher produces an upper-level action $a^u$ which corresponds to the environment parameters. These environment parameters are subsequently used to generate specific environment instances. The lower-level student policy $\pi$ will be trained on the generated environments for $C$ training steps. The upper-level teacher collects and stores the student policy evolution transition $(s^u, a^u, r^u, s^{u,\prime})$ every $C$ times steps for off-policy training. The teacher agent is trained to maximize the cumulative reward giving the budget for the number of generated environments. The choice of the evaluation environments will be discussed in Section 3.3.

**Lower-level student MDP.** The generated environment is fully specified for the student, characterized by a Partially Observable Markov Decision Process (POMDP), which is defined by a tuple $\mathcal{M}_{\vec{\theta}} = <A, O, S^{\vec{\theta}}, \mathcal{P}^{\vec{\theta}}, \mathcal{I}^{\vec{\theta}}, \mathcal{R}^{\vec{\theta}}, \gamma>$, where $A$ represents the set of actions, $O$ is the set of observations, $S^{\vec{\theta}}$ is the set of states determined by the environment parameters $\vec{\theta}$, similarly, $\mathcal{P}^{\vec{\theta}}$ is the environment-dependent transition function, and $\mathcal{I}^{\vec{\theta}} : \vec{\theta} \to O$ is the environment-dependent observation function, $\mathcal{R}^{\vec{\theta}}$ is the reward function, and $\gamma$ is the discount factor. At each time step $t$, the environment produces a state observation $s_t \in S^{\vec{\theta}}$, the student agent samples the action $a_t \sim A$ and interacts with environment $\vec{\theta}$. The environment yields a reward $r_t$ according to the reward function $\mathcal{R}^{\vec{\theta}}$. The student agent is trained to maximize their cumulative reward $V^{\vec{\theta}}(\pi) = \sum_{t=0}^{C} \gamma^t r_t$ for the current environment under a finite time horizon $C$. The student agent will learn a good general capability from training on a sequence of generated environments.

The hierarchical framework enables the teacher agent to systematically measure and enhance the general capability of the student agent and to adapt the training process accordingly. However, it's worth noting that collecting student policy evolution trajectories $(s^u, a^u, r^u, s^{u,'})$ to train the teacher agent is notably slow and resource-intensive, since each transition in the upper-level teacher MDP encompasses a training horizon of $C$ timesteps for the student in the generated environment. Thus, it is essential to reduce the need for costly collection of upper-level teacher experiences.

## 3.2 Generative Trajectory Modeling

In this section, we will formally introduce a generative model designed to ease the collection of upper-level MDP experience. This will allow us to train our teacher policy more efficiently. In particular, we first utilize a diffusion model to learn the conditional data distribution from the collected experiences $\tau = \{(s_t^u, a_t^u, r_t^u, s_t^{p,'})\}$. Later we can use the reverse chain in the diffusion model to generate the synthetic trajectories that can be used to help train the teacher agent, thereby alleviating the need for extensive and time-consuming collection of upper-level teacher experiences. We deal with two different types of timesteps in this section: one for the diffusion process and the other for the upper-level teacher agent, respectively. We use subscripts $k \in 1, \ldots, K$ to represent diffusion timesteps and subscripts $t \in 1, \ldots, T$ to represent trajectory timesteps in the teacher's experience.

In the image domain, the diffusion process is implemented across all pixel values of the image. In our setting, we diffuse over the next state $s^{u,'}$ conditioned the given state $s^u$ and action $a^u$. We construct our generative model according to the conditional diffusion process:

$$q(s_k^{u,'}|s_{k-1}^{u,'}), \quad p_\phi(s_{k-1}^{u,'}|s_k^{u,'}, s^u, a^u)$$

As usual, $q(s_k^{u,'}|s_{k-1}^{u,'})$ is the predefined forward noising process while $p_\phi(s_{k-1}^{u,'}|s_k^{u,'}, s^u, a^u)$ is the trainable reverse denoising process. We begin by randomly sampling the collected experiences $\tau = \{(s_t^u, a_t^u, r_t^u, s_t^{u,'})\}$ from the real experience buffer $\mathcal{B}_{real}$. Giving the observed state $s^u$ and action $a^u$, we use the reverse process $p_\phi$ to represent the generation of the next state $s^{u,'}$:

$$p_\phi(s_{0:K}^{u,'}|s^u, a^u) = \mathcal{N}(s_K^{u,'}; 0, \mathbf{I}) \prod_{k=1}^{K} p_\phi(s_{k-1}^{u,'}|s_k^{u,'}, s^u, a^u)$$

At the end of the reverse chain, the sample $s_0^{u,'}$, is the generated next state $s^{u,'}$. Similar to Ho et al. [6], we parameterize $p_\phi(s_{k-1}'|s_k', s^u, a^u)$ as a noise prediction model with the covariance matrix fixed as $\Sigma_\phi(s_k^{u,'}, s^u, a^u, k) = \beta_i \mathbf{I}$, and the mean is

$$\mu_\phi(s_i^{u,'}, s^u, a^u, k) = \frac{1}{\sqrt{\alpha_k}}\left(s_k^{u,'} - \frac{\beta_k}{\sqrt{1-\bar{\alpha}_k}}\epsilon_\phi(s_k^{u,'}, s^u, a^u, k)\right)$$

$\epsilon_\phi(s_k^{u,'}, s^u, a^u, k)$ is the trainable denoising function, which aims to estimate the noise $\epsilon$ in the noisy input $s_k^{u,'}$ at step $k$.

**Training objective.** We employ a similar simplified objective to train the conditional $\epsilon$- model:

$$\mathcal{L}(\phi) = \mathbb{E}_{(s^u, a^u, s^{u,'})\sim\tau, k\sim\mathcal{U}, \epsilon\sim\mathcal{N}(0,\mathbf{I})}\left[\|\epsilon - \epsilon_\phi(s_k^{u,'}, s^u, a^u, k)\|^2\right] \tag{6}$$

Where $s_k^{u,'} = \sqrt{\bar{\alpha}_k}s^{u,'} + \sqrt{1-\bar{\alpha}_k}\epsilon$. The intuition for the loss function $\mathcal{L}(\phi)$ is to predict the noise $\epsilon \sim \mathcal{N}(0, \mathbf{I})$ at the denoising step $k$, and the diffusion model is essentially learning the student policy involution trajectories collected in the real experience buffer $\mathcal{B}_{reals}$. Note that the reverse process necessitates a substantial number of steps $K$ [15]. Recent research by Xiao et al. [22] has demonstrated that enabling denoising with large steps can reduce the total number of denoising steps $K$. To expedite the relatively slow reverse sampling process (as it requires computing $\epsilon_\phi$ networks $K$ times), we use a small value of $K$. Similar to Wang et al. [20], while simultaneously setting $\beta_{\min} = 0.1$ and $\beta_{\max} = 10.0$, we define:

$$\beta_k = 1 - \exp\left(\beta_{\min} \times \frac{1}{K} - 0.5(\beta_{\max} - \beta_{\min})\frac{2k-1}{K^2}\right)$$

This noise schedule is derived from the variance-preserving Stochastic Differential Equation by Song et al. [16].

**Generate synthetic trajectories.**Once the diffusion model has been trained, it can be used to generate synthetic experience data by starting with a draw from the prior $s_K^{u,\prime} \sim \mathcal{N}(0, \mathbf{I})$ and successively generating denoised next state, conditioned on the given $s^u$ and $a^u$ through the reverse chain $p_\phi$. Note that the giving condition action $a$ can either be randomly sampled from the action space or use another diffusion model to learn the action distribution giving the initial state $s^u$. This new diffusion model is essentially a behavior-cloning model that aims to learn the teacher policy $\Lambda(a^u|s^u)$. This process is similar to the work of Wang et al. [20]. We discuss this process in detail in the appendix. In this paper, we randomly sample $a^u$ as it is straightforward and can also increase the diversity in the generated synthetic experience to help train a more robust teacher agent.

After obtaining the generated next state $s^{u,\prime}$ conditioned on $s^u, a^u$, we compute reward $r^u$ using teacher's reward function $R(s^u, a^u, s^{u,\prime})$. The specifics of how the reward function is chosen are explained in the following section.

### 3.3 Rewards and Choice of evaluate environments

**Selection of evaluation environments.** The upper-level teacher generates environments tailored for the lower-level student to improve its general capability. Thus it is important to select a set of diverse suitable evaluation environments as the performance vector reflects the student agent's general capabilities and serves as an approximation of the policy's embedding. Fontaine and Nikolaidis [5] propose the use of quality diversity (QD) optimization to collect high-quality environments that exhibit diversity for the agent behaviors. Similarly, Bhatt et al. [1] introduce a QD-based algorithm for dynamically designing such evaluation environments based on the current agent's behavior. However, it's worth noting that this QD-based approach can be tedious and time-consuming, and the collected evaluation environments heavily rely on the given agent policy.

Given these considerations, it is natural to take advantage of the domain randomization algorithm, as it has demonstrated compelling results in generating diverse environments and training generally capable agents. In our approach, we first discretize the environment parameters into different ranges, then randomly sample from these ranges, and combine these parameters to generate evaluation environments. This method can generate environments that may induce a diverse performance for the same policy, and it shows promising empirical results in the final experiments.

**Reward design.** We define the reward function for the upper-level teacher policy as a parameterized function based on the improvement in student performance in the evaluation environments after training in the generated environment:

$$R(s^u, a^u, s^{u,\prime}) = \sum_{i=1}^{m}(p_i' - p_i)$$

This reward function gives positive rewards to the upper-level teacher for taking action to create the right environment to improve the overall performance of students across diverse environments. However, it may encourage the teacher to obtain higher rewards by sacrificing student performance in one subset of evaluation environments to improve student performance in another subset, which conflicts with our objective to develop a student agent with general capabilities. Therefore, we need to consider fairness in the reward function to ensure that the generated environment can improve student's general capabilities. Similar to [4], we build our fairness metric on top of the change in student's performance in each evaluation environment, denoted as $\omega_i = p_i' - p_i$, and we have $\bar{\omega} = \frac{1}{m}\sum_{i=1}^{m}\omega_i$. We then measure the fairness of the teacher's action using the coefficient of variation of student performances:

$$cv(s^u, a^u, s^{u,\prime}) = \sqrt{\frac{1}{m-1}\sum_i \frac{(\omega_i - \bar{\omega})^2}{\bar{\omega}^2}} \tag{7}$$

A teacher is considered to be fair if and only if the $cv$ is smaller. As a result, our reward function is:

$$R(s^u, a^u, s^{u,\prime}) = \sum_{i=1}^{m}(p_i' - p_i) - \eta \cdot cv(s^u, a^u, s^{u,\prime}) \tag{8}$$

Here $\eta$ is the coefficient that balances the weight of fairness in the reward function (We set a small value to $\eta$). This reward function motivates the teacher to generate training environments that can improve student's general capability.

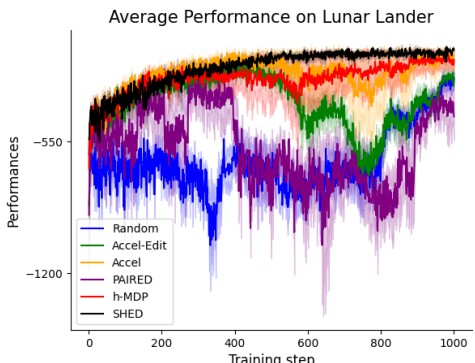 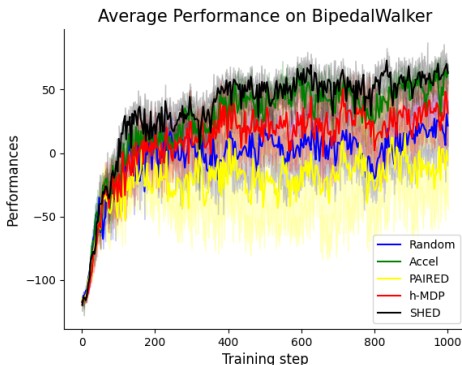

Figure 3: *Left*: The average zero-shot transfer performances on the test environments in the Lunar lander environment (mean and standard error). *Right*: The average zero-shot transfer performances on the test environments in the BipedalWalker (mean and standard error).

## 4 Experiments

In this section, we conduct experiments to compare *SHED* to other leading approaches on three domains: Lunar Lander, maze and a modified BipedalWalker environment. Experimental details and hyperparameters can be found in the Appendix. Specifically, our primary comparisons involve *SHED* and *h-MDP* (our proposed hierarchical approach without diffusion model aiding in training) against four baselines: domain randomization [19], ACCEL, [10], Edited ACCEL(with slight modifications that it does not revisit the previously generated environments), PAIRED [3]. In all cases, we train a student agent via Proximal Policy Optimization (PPO [13], and train the teacher agent via Deterministic policy gradient algorithms(DDPG [14]), because DDPG is an off-policy algorithm and can learn from both real experiences and the synthetic experiences.

**Setup.** For each domain, we construct a set of evaluation environments and a set of test environments. The vector of student performances in the evaluation environments is used as the approximation of the student policy (as the observation to teacher agent), and the performances in the test environments are used to represent the student's zero-shot transfer performances (general capabilities). Note that in order to obtain a fair comparison of zero-shot transfer performance, the evaluation environments and test environments do not share the same environment and they are not present during training.

**Lunar Lander.** This is a classic rocket trajectory optimization problem. In this domain, student agents are tasked with controlling a lander's engine to safely land the vehicle. Before the start of each episode, teacher algorithms determine the environment parameters that are used to generate environments in a given play-through, which includes gravity, wind power, and turbulence power. These parameters directly alter the difficulty of landing the vehicle safely. The state is an 8-dimensional vector, which includes the coordinates of the lander, its linear velocities, its angle, its angular velocity, and two booleans that represent whether each leg is in contact with the ground or not.

We train the student agent for 1e6 environment time steps and periodically test the agent in test environments. The parameters for the test environments are randomly generated and fixed during training. We report the experiment results on the left side of Figure 3. As we can see, student agents trained under *SHED* consistently outperform other baselines and have minimal variance in transfer performance. During training, the baselines, except h-MDP, show a performance dip in the middle. This phenomenon could potentially be attributed to the inherent challenge of designing the appropriate environment instance in the large environment parameter space. This further demonstrates the effectiveness of our hierarchical design (*SHED* and h-MDP), which can successfully create environments that are appropriate to the current skill level of the students.

**Bipedalwalker.** We also evaluate *SHED* in the modified BipedalWalker from Parker-Holder et al. [10]. In this domain, the student agent is required to control a bipedal vehicle and navigate across the terrain, and the student receives a 24-dimensional proprioceptive state with respect to its lidar sensors, angles, and contacts. The teacher is tasked to select eight variables (including ground roughness, the

number of stairs steps, min/max range of pit gap width, min/max range of stump height, and min/max range of stair height) to generate the corresponding terrain.

We use similar experiment settings in prior UED works, we train all the algorithms for 1e7 environment time steps, and then evaluate their generalization ability on ten distinct test environments in Bipedal-Walker domain. The parameters for the test environments are randomly generated and fixed during training. As shown in Figure 3, our proposed method *SHED* surpasses all other baselines and achieves performance levels nearly on par with the SOTA (ACCEL). Meanwhile, SHED maintains a slight edge in terms of stability and overall performance and PAIRED suffers from a considerable degree of variance in its performance.

**Partially observable Maze.**  Here we study navigation tasks, where an agent must explore to find a goal while navigating around obstacles. The environment is partially observable, and the agent's field of view is limited to a $3 \times 3$ grid area. Unlike the previously mentioned domains, maze environments are non-parametric and cannot be directly represented by compact parameter vectors due to their high complexity. To solve this challenge, we propose a novel method to generate maze by leveraging advances in large language models (e.g., ChatGPT). Specifically, we implement a retrieval-augmented generation (RAG) process to optimize the ChatGPT's output such that it can generate desired maze environments. This process ensures that large language models reference authoritative knowledge bases to generate feasible mazes. To simplify the teacher's action space, we extracted several key factors that constitute the teacher's action space (environmental parameters) for maze generation. Details on maze generation are provided in Appendix D.3, and prompt are included in Appendix D.4.

The average zero-shot transfer performances are reported in Figure 4. Notably, *SHED* demonstrates the highest performance, consistently improving and achieving the highest cumulative rewards. The performance of h-MDP steadily improves but does not reach the highest levels, which further highlights the advantages of incorporating the generated synthetic datasets to train an effective RL teacher agent. Meanwhile, Accel-Edit and Accel show higher variances in performance, indicating that random teachers are less stable in finding a suitable environment to train student agents.

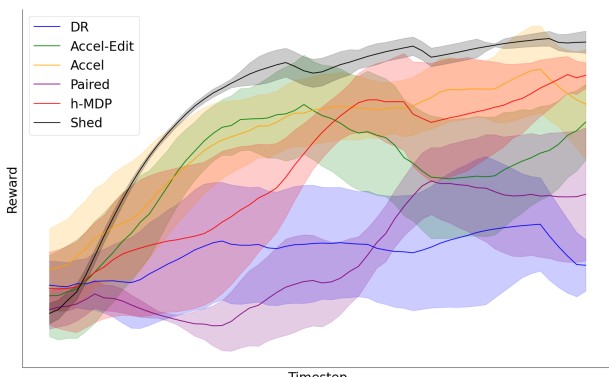

Figure 4: Average zero-shot transfer performance on the test environments in the maze environments.

**Ablation and additional Experiments**  In Appendix C, we evaluate the ability of the diffusion model to generate the synthetic student policy involution trajectories. We further provide ablation studies to assess the impact of different design choices in Appendix E.1. Additionally, in Appendix E.2, we conduct experiments to show how the algorithm performs under different settings, including scenarios with a larger budget constraint on the number of generated environments or a larger weight assigned to CV fairness rewards. Notably, all results consistently demonstrate the effectiveness of our approach.

# 5  Conclusion

In this paper, we introduce an adaptive approach for efficiently training a generally capable agent under resource constraints. Our approach is general, utilizing an upper-level MDP teacher agent that can guide the training of the lower-level MDP student agent agent. The hierarchical framework can incorporate techniques from existing UED works, such as prioritized level replay (revisiting environments with high learning potential). Furthermore, we have described a method to assist the experience collection for the teacher when it is trained in an off-policy manner. Our experiment demonstrates that our method outperforms existing UED methods, highlighting its effectiveness as a curriculum-based learning approach within the UED framework.

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

## A Theorem

**Theorem 1** *There exists a finite evaluation environment set that can capture the student's general capabilities and the performance vector $[p_1, \ldots, p_m]$ is a good representation of the student policy.*

To prove this, we first provide the following Assumption:

**Assumption 1** *Let $p(\pi, \vec{\theta})$ denote the performance of student policy $\pi$ in an environment $\vec{\theta}$. For $\forall i$-th dimension of the environment parameters, denoted as $\theta_i$, when changing the $\theta_i$ to $\theta_i'$ to get a new environment $\vec{\theta'}$ while keeping other environment parameters fixed, there $\exists \delta_i > 0$, if $|\theta_i' - \theta_i| \leq \delta_i$, we have $|p(\pi, \vec{\theta'}) - p(\pi, \vec{\theta})| \leq \epsilon_i$, where $\epsilon_i \to 0$.*

If this is true, we then can construct a finite set of environments, and the student performances in those environments can represent the performances in all potential environments generated within the certain environment parameters open interval combinations, and the set of those open intervals combinations cover the environment parameter space $\Theta$.

We begin from the simplest case where we only consider using one environment parameter to generate environments, denoted as $\theta_i$. We can construct a finite environment parameter set for environment parameters, which is $\{\theta_i^{min} + 1/2 * \delta_i, \theta_i^{min} + 3/2 * \delta_i, \theta_i^{min} + 7/2 * \delta_i, \ldots, \theta_i^{max} - \delta_i/2\}$. Assume the set size is $L_i$. We let the set $\{\vec{\theta_i}\}_{i=1}^{L_i}$ denote the corresponding generated environments. This is served as the **representative environment set**. Then the student performances in those environments are denoted as $\{p(\pi, \vec{\theta_i})\}_{i=1}^{L_i}$, which we call it as **representative performance vector set**. We can divide the space for $\theta_i$ into a finite set of open intervals with size $L_i$, which is $\{[\theta_i^{min}, \theta_i^{min} + 3/2 * \delta_i), (\theta_i^{min} + 1/2 * \delta_i, \theta_i^{min} + 5/2\delta_i), (\theta_i^{min} + 5/2 * \delta_i, \theta_i^{min} + 9/2 * \delta_i), \ldots, (\theta_i^{max} - 3/2 * \delta_i, \theta_i^{max}]\}$, which we call it as **representative parameter interval set**, also denoted as $\{(\theta_i - \delta, \theta_i + \delta)\}_{i=1}^{L_i}$. For any environment generated in those intervals, denoted as $\vec{\theta_i'}$, the performance $p(\pi, \vec{\theta_i'})$ can always be represented by the $p(\pi, \vec{\theta_i})$ which is in the same interval, as $|p(\pi, \vec{\theta_i'}) - p(\pi, \vec{\theta_i})| \leq \epsilon_i$, where $\epsilon_i \to 0$. In such cases, the finite set of environmental parameter intervals $\{\theta_i^{min} + 1/2 * \delta_i, \theta_i^{min} + 3/2 * \delta_i, \theta_i^{min} + 7/2 * \delta_i, \ldots, \theta_i^{max} - \delta_i/2\}$ fully covers the entire parameter space $\Theta$. We can find a representative environment set $\{\vec{\theta_i}\}_{i=1}^{L_i}$ that is capable of approximating the performance of the student policy within the open parameter intervals combination. This set effectively characterizes the general performance capabilities of the student policy $\pi$.

Then we extend to two environment parameter design space cases. Let's assume that the environment is generated by two-dimension environment parameters. Then, for each environment parameter, $\theta_i \in \{\theta_1, \theta_2\}$. We can find the same open interval set for each parameter. Specifically, for each $\theta_i$, there exists a $\delta_i$, such that if $|\theta_i' - \theta_i| \leq \delta_i$, we have $|p(\pi, \vec{\theta'}) - p(\pi, \vec{\theta})| \leq \epsilon_i$, where $\epsilon_i \to 0$. Hence, we let $\delta = \min\{\delta_1, \delta_2\}$ and $\epsilon = \epsilon_1 + \epsilon_2$. Thus the new **representative environment set** is the set that includes the any combination of $\{[\theta_1, \theta_2]\}$ where $\theta_1 \in \{\vec{\theta_i}\}_{i=1}^{L_1}$ and $\theta_2 \in \{\vec{\theta_j}\}_{j=1}^{L_2}$. We can get the **representative performance vector set** as $\{p(\pi, [\vec{\theta_i}, \vec{\theta_j}])\}_{i \in [1, L_1], j \in [1, L_2]}$. We then can construct the **representative parameter interval set** as $\{[(\theta_i - \delta, \theta_i + \delta), (\theta_j - \delta, \theta_j + \delta)]\}_{i \in [1, L_1], j \in [1, L_j]}$. As a result, for any new environments $[\vec{\theta_i'}, \vec{\theta_j'}]$, we can find the representative environment whose environment parameters are in the same parameter interval $[\vec{\theta_i}, \vec{\theta_j}]$, such that their performance difference is smaller than $\epsilon = \epsilon_1 + \epsilon_2$ for all $\forall i \in [1, L_1], \forall j \in [1, L_2]$:

$$
\begin{aligned}
|p(\pi, [\vec{\theta_i'}, \vec{\theta_j'}]) - p(\pi, [\vec{\theta_i}, \vec{\theta_j}])| &= |p(\pi, [\vec{\theta_i'}, \vec{\theta_j'}]) - p(\pi, [\vec{\theta_i'}, \vec{\theta_j}]) + p(\pi, [\vec{\theta_i'}, \vec{\theta_j}]) - p(\pi, [\vec{\theta_i}, \vec{\theta_j}])| \\
&\leq |p(\pi, [\vec{\theta_i'}, \vec{\theta_j'}]) - p(\pi, [\vec{\theta_i'}, \vec{\theta_j}])| + |p(\pi, [\vec{\theta_i'}, \vec{\theta_j}]) - p(\pi, [\vec{\theta_i}, \vec{\theta_j}])| \\
&\leq \delta_j + \delta_i \\
&= \delta
\end{aligned}
\tag{9}
$$

In such cases, the finite set of environmental parameter intervals $\{[(\theta_i - \delta, \theta_i + \delta), (\theta_j - \delta, \theta_j + \delta)]\}_{i \in [1, L_1], j \in [1, L_j]}$ fully covers the entire parameter space $\Theta$. We can find a representative environment set $\{\vec{\theta_i}\}_{i=1}^{L_i}$ that is capable of approximating the performance of the student policy within the

Table 1: The teacher policies corresponding to the three approaches for UED. $U(\Theta)$ is a uniform distribution over environment parameter space, $\tilde{D}_\pi$ is a baseline distribution, $\bar{\theta}_\pi$ is the trajectory which maximizes regret of $\pi$, and $v_\pi$ is the value above the baseline distribution that $\pi$ achieves on that trajectory, $c_\pi$ is the negative of the worst-case regret of $\pi$. Details are described in PAIRED [3].

| UED Approaches | Teacher Policy | Decision Rule |
|---|---|---|
| DR [19] | $\Lambda(\pi) = U(\Theta)$ | Randomly sample |
| PARIED [3] | $\Lambda(\pi) = \{\bar{\theta}_\pi : \frac{c_\pi}{v_\pi}, \tilde{D}_\pi : \text{otherwise}\}$ | Minimax Regret |
| SHED (ours) | $\Lambda(\pi) = \underset{\vec{\theta} \in \Theta}{\arg \max} Q_\pi(s = \pi, a = \vec{\theta})$ | Maximize reward |

open parameter intervals combination. This set effectively characterizes the general performance capabilities of the student policy $\pi$.

Similarly, we can show this still holds when the environment is constructed by a larger dimension environment parameters, where we set $\delta = \min\{\delta_i\}$, and $\epsilon = \sum_i \epsilon_i$, and we have $\delta > 0$, $\epsilon \to 0$. The overall logic is that we can find a finite set, which is called **representative environment set**, and we can use performances in this set to represent any performances in the environments generated in the **representative parameter interval set**, which is called **representative performance vector set**. Finally, we can show that **representative parameter interval set** fully covers the environment parameter space. Thus there exists a finite evaluation environment set that can capture the student's general capabilities and the performance vector, called **representative performance vector set**, $[p_1, \ldots, p_m]$ is a good representation of the student policy.

# B  Details about the Generative model

## B.1  Generative model to generate synthetic next state

Here, we describe how to leverage the diffusion model to learn the conditional data distribution in the collected experiences $\tau = \{(s_t^u, a_t^u, r_t^u, s_t^{u,\prime})\}$. Later we can use the trainable reverse chain in the diffusion model to generate the synthetic trajectories that can be used to help train the teacher agent, resulting in reducing the resource-intensive and time-consuming collection of upper-level teacher experiences. We deal with two different types of timesteps in this section: one for the diffusion process and the other for the upper-level teacher agent, respectively. We use subscripts $k \in 1, \ldots, K$ to represent diffusion timesteps and subscripts $t \in 1, \ldots, T$ to represent trajectory timesteps in the teacher's experience.

In the image domain, the diffusion process is implemented across all pixel values of the image. In our setting, we diffuse over the next state $s^{u,\prime}$ conditioned the given state $s^u$ and action $a^u$. We construct our generative model according to the conditional diffusion process:

$$q(s_k^{u,\prime}|s_{k-1}^{u,\prime}), \quad p_\phi(s_{k-1}^{u,\prime}|s_k^{u,\prime}, s^u, a^u)$$

As usual, $q(s_k^{u,\prime}|s_{k-1}^{u,\prime})$ is the predefined forward noising process while $p_\phi(s_{k-1}^{u,\prime}|s_k^{u,\prime}, s^u, a^u)$ is the trainable reverse denoising process. We begin by randomly sampling the collected experiences $\tau = \{(s_t^u, a_t^u, r_t^u, s_t^{u,\prime})\}$ from the real experience buffer $\mathcal{B}_{real}$.

We drop the superscript $u$ here for ease of explanation. Giving the observed state $s$ and action $a$, we use the reverse process $p_\phi$ to represent the generation of the next state $s'$:

$$p_\phi(s'_{0:K}|s, a) = \mathcal{N}(s'_K; 0, \mathbf{I}) \prod_{k=1}^{K} p_\phi(s'_{k-1}|s'_k, s, a) \tag{10}$$

At the end of the reverse chain, the sample $s'_0$, is the generated next state $s'$. As shown in Section 2.2, $p_\phi(s'_{k-1}|, s'_k, s, a)$ could be modeled as a Gaussian distribution $\mathcal{N}(s'_{k-1}; \mu_\theta(s'_k, s, a, k), \Sigma_\theta(s'_k, s, a, k))$. Similar to Ho et al. [6], we parameterize $p_\phi(s'_{k-1}|s'_k, s, a)$ as a noise prediction model with the covariance matrix fixed as

$$\Sigma_\theta(s'_k, s, a, k) = \beta_i \mathbf{I}$$

and mean is

$$\mu_\theta(s'_i, s, a, k) = \frac{1}{\sqrt{\alpha_k}} \left( s'_k - \frac{\beta_k}{\sqrt{1 - \bar{\alpha}_k}} \epsilon_\theta(s'_k, s, a, k) \right)$$

Where $\epsilon_\theta(s'_k, s, a, k)$ is the trainable denoising function, which aims to estimate the noise $\epsilon$ in the noisy input $s'_k$ at step $k$. Specifically, giving the sampled experience $(s, a, s')$, we begin by sampling $s'_K \sim \mathcal{N}(0, \mathbf{I})$ and then proceed with the reverse diffusion chain $p_\phi(s'_{k-1}|, s'_k, s, a)$ for $k = K, \ldots, 1$. The detailed expression for $s'_{k-1}$ is as follows:

$$\frac{s'_k}{\sqrt{\alpha_k}} - \frac{\beta_k}{\sqrt{\alpha_k(1 - \bar{\alpha}_k)}} \epsilon_\theta(s'_k, s, a, k) + \sqrt{\beta_k}\epsilon, \tag{11}$$

where $\epsilon \sim \mathcal{N}(0, \mathbf{I})$. Note that $\epsilon = 0$ when $k = 1$.

**Training objective.** We employ a similar simplified objective, as proposed by Ho et al. [6] to train the conditional $\epsilon$- model through the following process:

$$\mathcal{L}(\theta) = \mathbb{E}_{(s,a,s')\sim\tau, k\sim\mathcal{U}, \epsilon\sim\mathcal{N}(0,\mathbf{I})} \left[ \|\epsilon - \epsilon_\phi(s'_k, s, a, k)\|^2 \right] \tag{12}$$

Where $s'_k = \sqrt{\bar{\alpha}_k}s' + \sqrt{1 - \bar{\alpha}_k}\epsilon$. $\mathcal{U}$ represents a uniform distribution over the discrete set $\{1, \ldots, K\}$. The intuition for the loss function $\mathcal{L}(\theta)$ tries to predict the noise $\epsilon \sim \mathcal{N}(0, \mathbf{I})$ at the denoising step $k$, and the diffusion model is essentially learning the student policy involution trajectories collected in the real experience buffer $\mathcal{B}_{reals}$. Note that the reverse process necessitates a substantial number of steps $K$, as the Gaussian assumption holds true primarily under the condition of the infinitesimally limit of small denoising steps [15]. Recent research by Xiao et al. [22] has demonstrated that enabling denoising with large steps can reduce the total number of denoising steps $K$. To expedite the relatively slow reverse sampling process outlined in Equation 3.2 (as it requires computing $\epsilon_\phi$ networks $K$ times), we use a small value of $K$, while simultaneously setting $\beta_{\min} = 0.1$ and $\beta_{\max} = 10.0$. Similar to Wang et al. [20], we define:

$$\beta_k = 1 - \alpha_k$$
$$= 1 - \exp\left(\beta_{\min} \times \frac{1}{K} - 0.5(\beta_{\max} - \beta_{\min})\frac{2k - 1}{K^2}\right)$$

This noise schedule is derived from the variance-preserving Stochastic Differential Equation by Song et al. [16].

**Generate synthetic trajectories.** Once the diffusion model has been trained, it can be used to generate synthetic experience data by starting with a draw from the prior $s'_K \sim \mathcal{N}(0, \mathbf{I})$ and successively generating denoised next state, conditioned on the given $s$ and $a$ through the reverse chain $p_\phi$ in Equation 3.2. Note that the giving condition action $a$ can either be randomly sampled from the action space (which is also the environment parameter space) or use another diffusion model to learn the action distribution giving the initial state $s$. In such case, this new diffusion model is essentially a behavior-cloning model that aims to learn the teacher policy $\Lambda(a|s)$. This process is similar to the work of Wang et al. [20]. We discuss this process in detail in the appendix. In this paper, we randomly sample $a$ as it is straightforward and can also increase the diversity in the generated synthetic experience to help train a more robust teacher agent.

### B.2 Generative model to generate synthetic action

Once the diffusion model has been trained, it can be used to generate synthetic experience data by starting with a draw from the prior $s'_K \sim \mathcal{N}(0, \mathbf{I})$ and successively generating denoised next state, conditioned on the given $s$ and $a$ through the reverse chain $p_\phi$ in Equation 3.2. Note that the giving condition action $a$ can either be randomly sampled from the action space (which is also the environment parameter space) or we can train another diffusion model to learn the action distribution giving the initial state $s$, and then use the trained new diffusion model to sample the action $a$ giving the state $s$. This process is similar to the work of Wang et al. [20].

In particular, We construct another conditional diffusion model as:

$$q(a_k|a_{k-1}), \quad p_\phi(a_{k-1}|a_k, s)$$

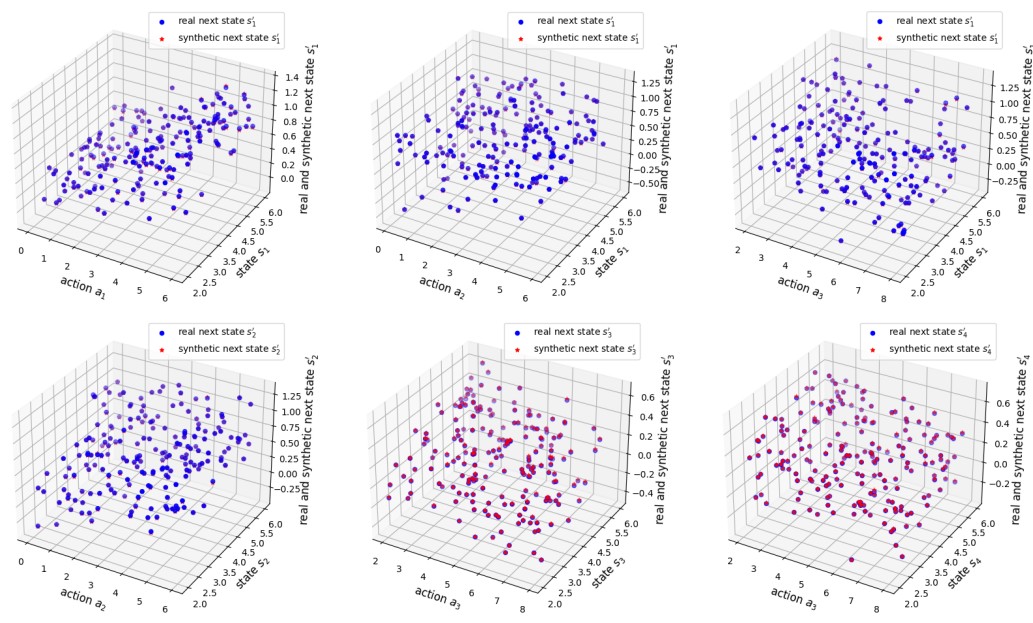

Figure 5: The distribution of the real $s'$ and the synthetic $s'$ conditioned on $(s, a)$.

As usual, $q(a_k|a_{k-1})$ is the predefined forward noising process while $p_\phi(a_{k-1}|a_k, s)$ is the trainable reverse denoising process. we represent the action generation process via the reverse chain of the conditional diffusion model as

$$p_\phi(a_{0:K}|s) = \mathcal{N}(a_K; 0, \mathbf{I}) \prod_{k=1}^{K} p_\phi(a_{k-1}|a_k, s) \tag{13}$$

At the end of the reverse chain, the sample $a_0$, is the generated action $a$ for the giving state $s$. Similarly, we parameterize $p_\phi(a_{k-1}|a_k, s)$ as a noise prediction model with the covariance matrix fixed as

$$\Sigma_\theta(a_k, s, k) = \beta_i \mathbf{I}$$

and mean is

$$\mu_\theta(a_i, s, k) = \frac{1}{\sqrt{\alpha_k}} \left( a_k - \frac{\beta_k}{\sqrt{1 - \bar{\alpha}_k}} \epsilon_\theta(a_k, s, k) \right)$$

Similarly, the simplified loss function is

$$\mathcal{L}^a(\theta) = \mathbb{E}_{(s,a)\sim\tau, k\sim\mathcal{U}, \epsilon\sim\mathcal{N}(0,\mathbf{I})} \left[ \|\epsilon - \epsilon_\phi(a_k, s, k)\|^2 \right] \tag{14}$$

Where $a_k = \sqrt{\bar{\alpha}_k}a + \sqrt{1 - \bar{\alpha}_k}\epsilon$. $\mathcal{U}$ represents a uniform distribution over the discrete set $\{1, \ldots, K\}$. The intuition for the loss function $\mathcal{L}^a(\theta)$ tries to predict the noise $\epsilon \sim \mathcal{N}(0, \mathbf{I})$ at the denoising step $k$, and the diffusion model is essentially a behavior cloning model to learn the student policy collected in the real experience buffer $\mathcal{B}_{reals}$.

Once this new diffusion model is trained, the generation of the synthetic experience can be formulated as:

- we first randomly sample the state from the collected real trajectories $s \sim \tau$;

- we use the new diffusion model discussed above to mimic the teacher's policy to generate the actions $a$;

- giving the state $s$ and action $a$, we use the first diffusion model presented in the main paper to generate the next state $s'$;

- we compute the reward $r$ according to the reward function, and add the final generated synthetic experience $(s, a, r, s')$ to the synthetic experience buffer $\mathcal{B}_{syn}$ to help train the teacher agent.

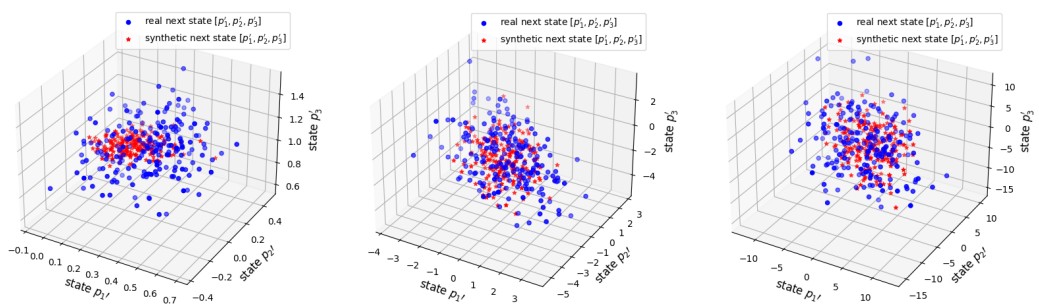

Figure 6: The distribution of the real $[s'_1, s'_2, s'_3]$(red) and the synthetic $[s'_1, s'_2, s'_3]$(blue) giving the fixed $(s^u, a^u)$. Specifically, the noise $\varepsilon$ in $f(s^u, a^u)$ is (i).*left* figure: $\varepsilon = \epsilon$, (ii).*middle* figure: $\varepsilon = 3 * \epsilon$, (iii).*right* figure: $\varepsilon = 10 * \epsilon$, where $\epsilon \sim \mathcal{N}(0, 1)$.

## C  Empirical analysis of generative model

### C.1  Ability to generate good synthetic trajectories

We begin by investigating *SHED*'s ability to assist in collecting experiences for the upper-level MDP teacher. This involves the necessity for *SHED* to prove its ability to accurately generate synthetic experiences for teacher agents. To check the quality of these generated synthetic experiences, we employ a diffusion model to simulate some data for validation (even though Diffusion models have demonstrated remarkable success across vision and NLP tasks).

We design the following experiment: given the teacher's observed state $s^u = [p_1, p_2, p_3, p_4, p_5]$, where $p_i$ denotes the student performance on $i$-th evaluation environment. and given the teacher's action $a^u = [a_1, a_2, a_3]$, which is the environment parameters and are used to generate corresponding environment instances. We use a neural network $f(s^u, a^u)$ to mimic the involution trajectories of the student policy $\pi$. That is, with the input of the state $s^u$ and action $a^u$ into the neural network, it outputs the next observed state $s^{u,'} = [p'_1, p'_2, p'_3, p'_4, p'_5]$, indicating the updated student performance vector on the evaluation environments after training in the environment generated by $a^u$. In particular, we add a noise $\varepsilon$ into $s^{u,'}$ to represent the uncertainty in the transition. We first train our diffusion model on the real dataset $(s^u, a^u, s^{u,'})$ generated by neural network $f(s^u, a^u)$. We then set a fixed $(s^u, a^u)$ pair and input them into $f(s^u, a^u)$ to generate 200 samples of real $s^{u,'}$. The trained diffusion model is then used to generate 200 synthetic $s^{u,'}$ conditioned on the fixed $(s^u, a^u)$ pair.

The results are presented in Figure 6, we can see that the generative model can effectively capture the distribution of real experience even if there is a large uncertainty in the transition, indicated by the value of $\varepsilon$. This provides evidence that the diffusion model can generate useful experiences conditioned on $(s^u, a^u)$. It is important to note that the marginal distribution derived from the reverse diffusion chain provides an implicit, expressive distribution, such distribution has the capability to capture complex distribution properties, including skewness and multi-modality.

### C.2  addition experiments on diffusion model

We further provide more results to show the ability of our generative model to generate synthetic trajectories where the noise is extremely small. In such cases, the actual next state $s'$ will converge to a certain value, and the synthetic next state $s^{syn,'}$ generated by the diffusion model should also be very close to that value, then the diffusion model has the ability to sample the next state $s_0^{syn,'}$ which can accurately represent the next state. We present the results in Figure 5. Specifically, this figure shows when the noise is very small in the actual next state, which is $0.05 * \epsilon$, and $\epsilon \sim \mathcal{N}(0, 1)$. Giving any condition $(s, a)$ pair, we selectively report on $(s_i, a_i)$, where $x$-axis is the $a_i$ value, and $y$-axis is the $s_i$ value. The student policy with initial performance vector $s$ is trained on the environments generated by the teacher's action $a$. We report the new performance $s'_i$ of student policy on $i$-th environments after training in the $z$-axis. In particular, if two points $s'_i$ and $s_i^{syn,'}$ are close, it indicates that the diffusion model can successfully generate the actual next state. As we can see, when the noise is extremely small, our diffusion model can accurately predict the next state of $s'_i$ giving any condition $(s, a)$ pair.

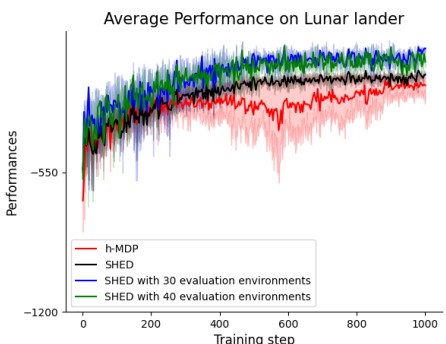
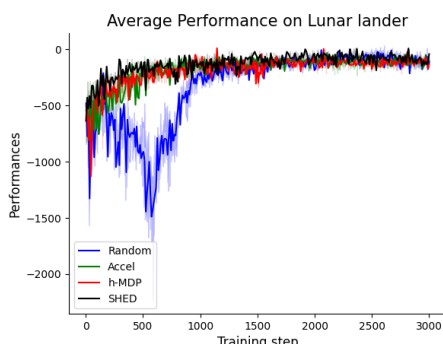

Figure 7: *Left*: The ablation study in the Lunar lander environment which investigates the effect of the size of the evaluation environment set. We provide the average zero-shot transfer performances on the test environments (mean and standard error). *Right*: Zero-shot transfer performance on the test environments under a longer time horizon in Lunar lander environments(mean and standard error).

# D   Additional Experiment Details

## D.1   Hyperparameters

We set the learning rate $1e-3$ for actor, and $3e-3$ for critic, we set gamma $\gamma = 0.999$, $\lambda = 0.95$, and set coefficient for the entropy bonus (to encourage exploration) as $0.01$. For each environment, we conduct 50 PPO updates for the student agent, and We can train on up to 50 environments, including replay. For our diffusion model, the diffusion discount is 0.99, and batch size is 64, $\tau$ is 0.005, learning rate is $3e-4$. The synthetic buffer size is $1000$, and the ratio is 0.25.

## D.2   Experiments Compute Resources

All the models were trained on a single NVIDIA GeForce RTX 3090 GPU and 16 CPUs.

## D.3   Maze document

Here we provide the document shows the instruction to generate feasible maze environments.

```
There are several factors that can affect the difficulty of a maze. Here are
some key factors to consider:
1. Maze Size: Larger mazes generally increase the complexity and difficulty
as the agent has more states to explore. Typically, the maze size should be
larger than 4x4 and smaller than 15*15.
- If the size is 7*7 or smaller, the maze size is considered easy.
- If the size is larger than 7*7 but smaller than 10*10, the maze size is
considered medium.
- If the maze size is larger than 10x10 but smaller than 15*15, the maze
size is considered hard.
2. Maze Structure: The complexity of the paths, including the number of twists,
turns, and dead-ends, can significantly impact navigation strategies. The
presence of narrow corridors versus wide-open spaces also plays a role.
- If there are fewer than 2 turns in the feasible path from the start position
to the end position, the maze structure is considered easy.
- If there are more than 2 turns but fewer than 4 turns in the path from the
start position to the end position, the maze structure is considered medium.
- If there are 4 or more turns in the path from the start position to the end
position, the maze structure is considered hard.
3. Goal Location: The distance from the starting position to the end position
also affects difficulty.
- If the path from the start position to the end position requires fewer than
```

632  5 steps, the goal location is considered easy.
633 - If the path from the start position to the end position requires 5 to 10
634  steps, the goal location is considered medium.
635 - If the path from the start position to the end position requires more than
636  10 steps, the goal location is considered hard.
637 4. Start Location: The starting position can also affect the difficulty of
638  the maze. The starting position is categorized into five levels:
639 - If the start position is close to 1, it means it should be located as close
640  to the top left of the maze.
641 - If the start position is close to 2, it means it should be located as close
642  to the top right of the maze.
643 - If the start position is close to 3, it means it should be located as close
644  to the bottom left of the maze.
645 - If the start position is close to 4, it means it should be located as close
646  to the bottom right of the maze.
647 - If the start position is close to 5, it means it should be located as close
648  to the center of the maze.
649 Please note that the generated maze uses -1 to represent blocks, 0 to
650 represent the feasible path, 1 to represent the start position, and 2 to represent
651 the end position. Must ensure that there is a feasible path in the generated maze!
652 A feasible path means that 1 and 2 are connected directly through 0s, or 1 and 2
653 are connected directly. For example:
654 Feasible Maze:
655 Maze = [
656   [0, -1, -1, 2],
657   [1, -1, 0, 0],
658   [0, -1, 0, -1],
659   [0, 0, 0, -1],
660 ]
661 Non-Feasible Mazes:
662 Maze = [
663   [0, -1, -1, 2],
664   [1, -1, 0, 0],
665   [0, -1, -1, 0],
666   [0, 0, 0, -1],
667 ]
668 Or
669 Maze = [
670   [1, -1],
671   [-1, 2]
672 ]
673 These second example does not have any feasible path.
674
675

### D.4   Prompt for RAG

677 We provide our prompt for the Retrieval Augmented Generation as follows:

678 Please refer to the document, and generate a maze with feasible path. The
679 difficulty level for the maze size is {maze_size_level}, and the difficulty
680 level for the maze structure is {maze_structure_level}, he difficulty level
681 for the goal location is {goal_location_level}, he difficulty level for
682 the start location is {start_position_level}.

 # E    Additional experiments

## E.1    Additional experiments about ablation studies

We also provide ablation analysis to evaluate the impact of different design choices in Lunar lander domain, including (a) a larger evaluation environment set; (b) a bigger budget for constraint on the number of generated environments (which incurs a longer training time horizon). The results are reported in Figure 7.

We explore the impact of introducing the diffusion model in collecting synthetic teacher's experience and varying the size of the evaluation environment set. Specifically, as we can see from the right side of Figure 7, the *SHED* consistently outperforms h-MDP, indicating the effectiveness of introducing the generative model to help train the upper-level teacher policy. Furthermore, we find that when increasing the size of the evaluation environment set, we can have a better result in the student transfer performances. The intuition is that a larger evaluation environment set, encompassing a more diverse range of environments, provides a better approximation of the student policy according to the Theorem 1. However, the reason why *SHED* with 30 evaluation environments slightly outperforms *SHED* with 40 evaluation environments is perhaps attributed to the increase in the dimension of the student performance vector, which amplifies the challenge of training an effective diffusion model with a limited dataset.

We conduct experiments in Lunar lander under a longer time horizon. The results are provided on the right side of Figure 7. As we can see, our proposed algorithm *SHED* can efficiently train the student agent to achieve the general capability in a shorter time horizon, This observation indicates that our proposed environment generation process can better generate the suitable environments for the current student policy, thereby enhancing its general capability, especially when there is a constraint on the number of generated environments.

## E.2    Additional experiments on Lunar lander

we also conduct experiments to show how the algorithm performs under different settings, such as a larger weight of cv fairness rewards ($\eta = 10$). The results are provided in Figure 8. We noticed an interesting finding: when fairness reward has a high weightage, our algorithm tends to generate environments at the onset that lead to a rapid decline and subsequent improvement in student performance across all test environments. This is done to avoid acquiring a substantial negative fairness reward and thereby maximize the teacher's cumulative reward. Notably, the student's final performance still surpasses other baselines at the end of training.

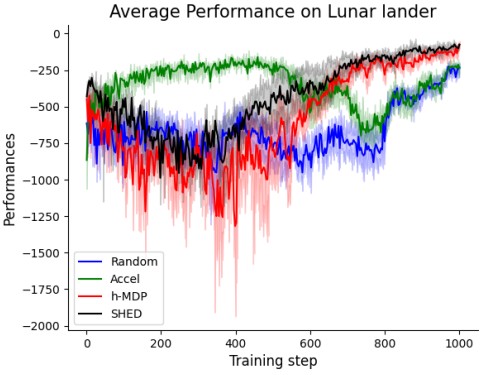

Figure 8: Zero-shot transfer performance on the test environments with a larger $cv$ value coefficient in Lunar lander environments.

We further show in detail how the performance of different methods changes in each testing environment during training (see Figure 9 and Figure 10 ).

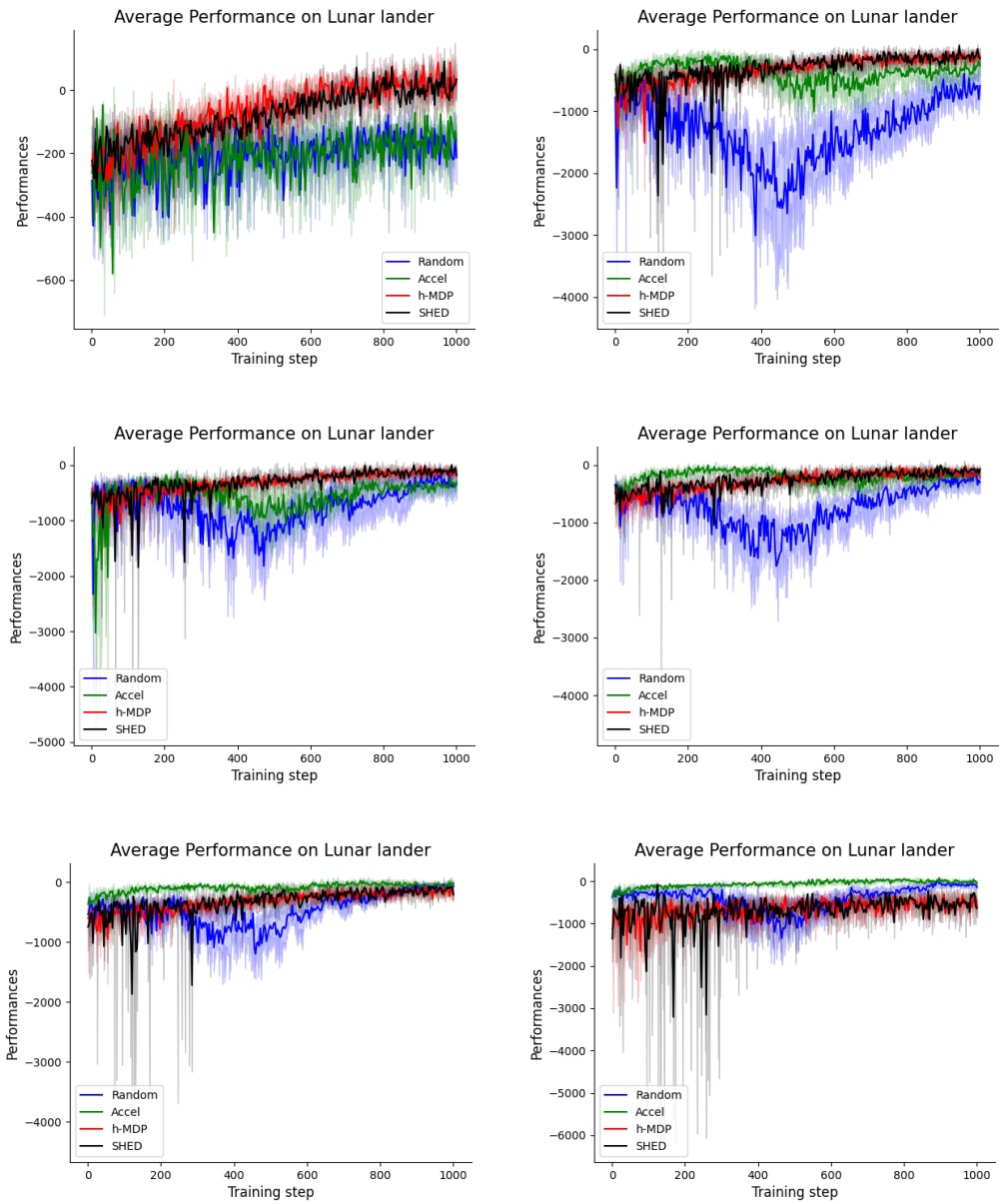

Figure 9: Detail how the performance of different methods changes in each testing environment during training (mean and error)

### E.3 Additional experiments on Maze

We selectively report some results of zero-shot transfer performances in maze environments. The results are provided in Figure

# F Discussion

## F.1 Limitations

The limitation of this work comes from the UED framework, as UED is limited to the use of parameterized environments. This results in our experimental domain being relatively simple.

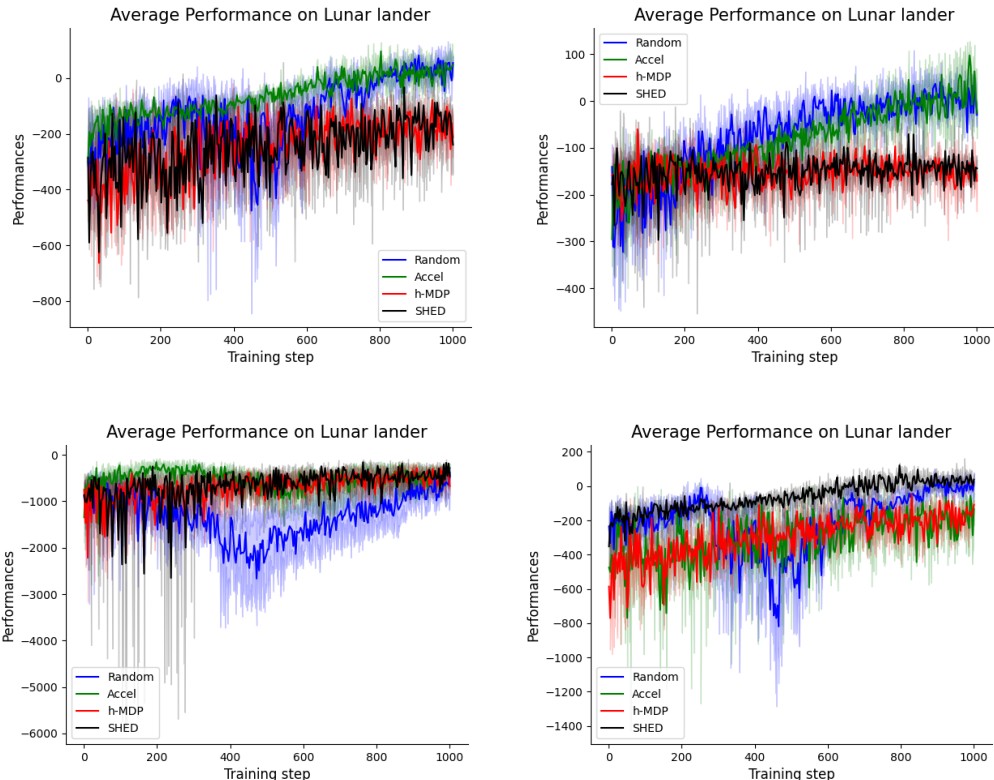

Figure 10: Detail how the performance of different methods changes in each testing environment during training (mean and error)

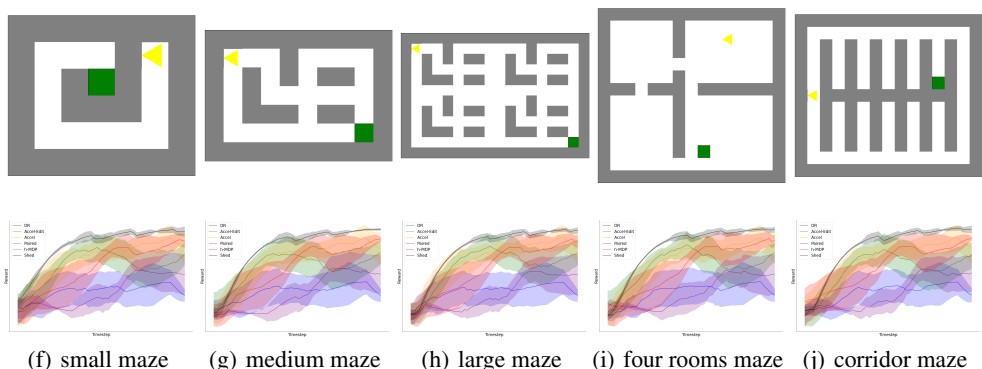

| (f) small maze | (g) medium maze | (h) large maze | (i) four rooms maze | (j) corridor maze |

Figure 11: Zeros-shot transfer performance on test environments in maze environemnts

However, our work proposes a new hierarchical structure, and our policy representation is not only of great help for UED, but also has certain inspirations for hierarchical RL. Additionally, in the world model of UED (Genie [2]), the environment generator (teacher) focuses on creating video games, a domain that is compatible with our proposed application of upsampling the teacher agent's experience using a diffusion model (since the state is image-based).

