# OpenReview forum: "Enhancing the Hierarchical Environment Design via Generative Trajectory Modeling"
_NeurIPS.cc/2024/Conference — Submitted to NeurIPS 2024_

### Official Review · Reviewer_8F47 · 2024-06-27

**Soundness:** 3
**Presentation:** 2
**Contribution:** 3
**Rating:** 6
**Confidence:** 4

**Summary:**

After rebuttal

While I think there are still some problems with this paper, e.g. the short training duration, and the slight exaggeration of claims (that SHED outperforms UED). I think, however, that the idea is nice, and getting RL environment design to work better is a good goal.


-----


This paper aims to improve Unsupervised Environment Design in two ways.
First, it introduces a hierarchical MDP formulation, where the top level corresponds to the teacher, and the lower level corresponds to the learning agent. Each transition in the top-level MDP involves training the lower level agent on generated levels. Related to this, they develop a state representation for the adversary, which is the performance of the agent on a fixed set of diverse levels.

Separately to this, they use a diffusion model to upsample the number of experiences for the teacher, effectively training on synthetic data, to improve sample efficiency.

**Strengths:**

- I think the H-MDP formulation itself is very valuable; it moves away from treating the generation of environments as a black-box, sparse reward, multi-step generation process (as in PAIRED), and towards a more informed process, where the teacher gets feedback in terms of the state (i.e., the performance vector).
- The analysis in the appendix investigating the ability of the model to generate good synthetic trajectories is useful.

**Weaknesses:**

- Major
	- The results do not very convincingly demonstrate that SHED is better than current SoTA. Looking at figure 3 particularly, I would say that ACCEL has about the same performance as SHED. However, comparing against the RL baseline, SHED does do better.
	- The method is limited in the types of environments it can generate. For instance, mazes are generated using an LLM instead of directly placing blocks. This method therefore is not quite as broad in scope as PAIRED or ACCEL, which can generate arbitrary environments.
	- Relatedly, in the minigrid experiments, do all methods generate the levels in the same way using an LLM, providing the difficulty numbers? It would be good to compare this against the standard version of ACCEL that directly places blocks in the maze level, as it does not have the same restriction as SHED.
- Minor
	- The figures can be improved:
		- Make the alpha value on the error bars a bit less
		- Keep colours consistent across figures, so the same method has the same colour
		- Keep capitalisation consistent across the figure labels.
	- line 80, the period ends on the line after the maths, it should end on the same line.
	- Footnote 1: Jiang et al. (2021) use (amongst others) the positive value loss, which is not quite the GAE, as it clips it at zero before summing.
	- equation one, you use $\beta_t$ but $t$ does not seem to be defined? Should this be $\beta_k$?
	- Line 159, PARIED should be PAIRED
	- There is no reward scale in figure 4
	- Figure 9's caption can be made clearer. I understand it to be the performance of each method in different testing environments.
	- Line 718 does not link to a figure.
	- Figure 11's caption: zero-shot and not zeros-shot
	- Capitalise the first word in the title of appendix C.2
	- In terms of notation, in line 96, $\pi^*$ usually has a dependence on $\theta$ (e.g. $\pi^*_\theta$) to indicate it is optimal w.r.t. that particular level.
	- Line 217, maybe add a citation to the first sentence, as I thought that is what you do, which confused me for a second.
	- line 237 space after period.
	- Line 241 "given" instead of giving?
	- Lines 296 - 297 are a bit confusing, as the word environment is used three times.
	- The assumption in theorem 1 is pretty strong.

**Questions:**

- It seems to me that the upsampling is very similar to [1], could you comment on that/cite the paper if indeed the technique is similar?
- Have you tried/have any intuition of what the effect would be if you set $\eta=0$?
- Furthermore, it seems like the specific evaluation environments you use can have a large impact. Have you looked at how variable results are when using different (still randomly generated) environment parameters. Relatedly, what happens if you just generate a few DR levels *without* explicitly discretising the environment parameters?
- Are you using the high entropy version of PAIRED introduced by [2]?
- Do you have an indication of how easy it would be to generate larger environments, with much larger parameter vectors?
- Do you think it would be possible to incorporate the multi-step generation of PAIRED into this process? E.g., still generate a maze step-by-step but also have the H-MDP structure and policy performance vector?
- Why can't you sample the action in the diffusion process from the teacher's policy?
- A reward of 50 in bipedal walker seems very low, could you compare against other papers / explain why it is low? E.g. [2] had rewards around 150


[1] Lu, Cong, et al. "Synthetic experience replay." _Advances in Neural Information Processing Systems_ 36 (2024).

[2] Mediratta, Ishita, et al. "Stabilizing unsupervised environment design with a learned adversary." _Conference on Lifelong Learning Agents_. PMLR, 2023.

**Limitations:**

I think the authors can list a few more limitations.
Primarily, the restriction on the type of environment that can be generated, i.e., it needs numerical parameters, and generating a maze outright is challenging. This is quite a large difference to prior settings.

---

> ### Author Rebuttal · Authors · 2024-08-06
>
> We thank the reviewer for valuable time and feedback,
>
> - **W1.** The results.
>
>   **A1.** It is crucial to note that SHED demonstrates more consistent and superior performance in the more complex BipedalWalker and Maze, as depicted in Figure 3,4. This suggests that SHED offers better generalizability and robustness across different environments. Moreover, the less variability in performance with SHED underscores its stability and reliability, which are essential qualities in real-world applications where consistent performance is critical.
>
> - **W2.** The method is limited to the types of environments.
>
>   **A2.** In the maze domain, learning block placement is challenging due to the sequential decision process involved (long horizon and large action space). Random placing blocks (ACCEL) may be viable, as it could result in mazes without solutions—especially under constraints of limited resources. To overcome these challenges, we introduced the use of LLM to manage the maze generation process on a more granular level. This approach not only mitigates the complexities associated with direct block placement but also introduces a degree of variability in the environment generation process. This innovative approach, which is unprecedented in the field of UED research, not only ensures the generation of solvable mazes but also extends the control in environment design. This environment design method has received positive feedback from reviewer EFDP.
>
>
> - **W3.** Do all methods use an LLM?
>
>   **A3.** Yes, in the maze domain, all methods generate mazes using ChatGPT, where we provide several key factors—rather than just single difficulty numbers—that define the teacher’s action space (environment parameters) for maze generation (see Appendix D). We can adjust the 'temperature' value in ChatGPT to enhance the diversity of the environments produced. This capability highlights the flexibility and robustness of our LLM-based method, offering an improvement over ACCEL that directly places blocks, which might generate unsolvable mazes and cannot capture the nuanced intentions for the desired generated mazes.
>
> - **Q1.** similar to [1]
>
>   **A1.** Thank you for pointing out this paper, which shares some similar motivations with our work. However, there are some key differences to note:
> 1. While [1] also adopts a diffusion model to generate trajectories, it lacks specific technical details on how these trajectories are generated. In contrast, our paper thoroughly describes the generation process.
> 2. Additionally, [1] focuses on upsampling student trajectories, whereas our proposed hierarchical MDP framework emphasizes the challenge of collecting the teacher's experience. Our method specifically upsamples the teacher's experience to improve training efficiency for the teacher.
>
> - **Q2.** intuition of $\eta=0$?
>
>   **A2.** As indicated in lines 270-274, when $\eta=0$, the teacher might obtain higher rewards by sacrificing student performance in one subset of evaluation environments to enhance it in another. This contradicts our objective of developing a student agent with general capabilities across all environments.
>
> We have included experiments in the appendix where $\eta$ is set to higher values, as shown in Fig. 8. We observed that the test performance deteriorates in certain environments.
>
> - **Q3.** How variable results are when using different environment parameters.
>
>   **A3.** In Appendix E, we provided details on how the number of evaluation environments influences the results. We demonstrate that increasing the size of the evaluation environment set leads to improved transfer performances.
>
> The variability in results is minimal when using different (randomly generated) environment parameters. This stability is due to our explicit discretization of the environment parameters, which ensures that the evaluation environments are sufficiently diverse and thus effectively reflect the student's general capabilities.
>
> However, when using just a few DR levels without explicitly discretizing the environment parameters for evaluation, the performance can be enhanced if the parameters are reasonable and do not heavily feature an OOD environment, as shown on the right in Fig. 3 in the attached PDF. Conversely, if similar evaluation environments predominantly include complex and unreasonable parameters (OOD), the results are notably poorer, as shown on the left in Fig. 3 in the PDF. Therefore, introducing diversity in the evaluation environments is essential to ensure stability in the overall performance.
>
> - **Q4.** high entropy version of PAIRED introduced by [2].
>
>   **A4.** We didn't use the entropy version of PAIRED, we will include this in the related work.
>
> - **Q5.** How easy it is to generate larger environments?
>
>   **A5.** In our framework, scaling up to generate larger environments with more extensive parameter vectors is feasible. Our framework is designed to be modular, allowing for a straightforward extension of the parameter space. However, the complexity and computational demands will increase with the size of the environment parameters.
>
> - **Q6.** Incorporate the multi-step generation of PAIRED?
>
>   **A6.** It is possible to integrate the multi-step generation approach of PAIRED into our framework; however, as mentioned in W2, learning to generate a maze step-by-step is inherently complex. In PAIRED, this process required hundreds of thousands of updates to learn effectively, which is feasible under settings with limited resources.
>
> - **Q7.** Sample the action in the diffusion process from the teacher's policy?
>
>   **A7.** Yes, sampling actions from the teacher's policy during the diffusion process is feasible.
>
> - **Q8.** results
>
>   **A8.** Please refer to A2 to Reviewer AEHi
>
> - **L1.** limitation
>
>   **A1.** While we have outlined some limitations in the appendix, we will revise and expand this to provide a more comprehensive analysis of potential weaknesses and areas for future improvement.

---

> > ### Comment · Reviewer_8F47 · 2024-08-07
> > **Discussion**
> >
> > Thank you for your response and your additional experiments.
> > I have some follow up questions and points to discuss
> >
> > > A1. It is crucial to note that SHED demonstrates more consistent and superior performance in the more complex BipedalWalker and Maze, as depicted in Figure 3,4. This suggests that SHED offers better generalizability and robustness across different environments. Moreover, the less variability in performance with SHED underscores its stability and reliability, which are essential qualities in real-world applications where consistent performance is critical.
> >
> >
> > I am not sure I am convinced. SHED performs about the same as ACCEL on lunar lander and bipedalwalker. Shed does slightly better in maze, but that is using a nonstandard way of generating mazes.
> >
> > > A7. Yes, sampling actions from the teacher's policy during the diffusion process is feasible.
> >
> > Then why did you not do this?
> >
> > > A8. Please refer to A2 to Reviewer AEHi
> >
> > Could you at least run evaluation on the same levels that ACCEL used? What is the computational demand of running Bipedal Walker for 30K PPO updates? (or at least something like 5k-10k, 50 doesn't seem enough to learn much)
> >
> > > A3
> >
> > Thanks for that, it is interesting.
> >
> > > A1. Thank you for pointing out this paper, which shares some similar motivations with our work. However, there are some key differences to note:
> > > While [1] also adopts a diffusion model to generate trajectories, it lacks specific technical details on how these trajectories are generated. In contrast, our paper thoroughly describes the generation process.
> >
> > To me the paper seems pretty clear, what details are lacking?
> >
> > > Additionally, [1] focuses on upsampling student trajectories, whereas our proposed hierarchical MDP framework emphasizes the challenge of collecting the teacher's experience. Our method specifically upsamples the teacher's experience to improve training efficiency for the teacher.
> >
> > The SynthER method is a way to upsample data for RL agents, and your teacher is still an RL agent. So I would still argue that it is very similar to what you are doing?
> >
> >
> > > A2. In the maze domain, learning block placement is challenging due to the sequential decision process involved
> >
> > Could you run a baseline comparing how well normal ACCEL does that places the blocks directly?
> >
> > Also please rectify all of the minor points in your revised manuscript.

---

> ### Author Response · Authors · 2024-08-10
> **Rebuttal by authors (1)**
>
> Thank you for your interest in our work and your prompt response. We have done additional experiments to address your concerns.
>
> **Q1.** SHED performs about the same as ACCEL on lunar lander and bipedalwalker. Shed does slightly better in maze, but that is using a nonstandard way of generating mazes.
>
> **A1.** We conducted experiments in the LunarLander environment to exclude very challenging test environments when testing zero-shot transfer performance. This is because all approaches perform poorly in these challenging environments, which would narrow the performance gap among them. Because we cannot directly show detailed results, we roughly report the results: After removing the challenging environments and completing 1,000 PPO updates, the average performance of SHED, h-MDP, and ACCEL improved significantly on LunarLander. However, ACCEL remains only slightly weaker than SHED. Here are the results: SHED: -6.019, h-MDP: -12.92, ACCEL: -8.706, Random: -30.65, PAIRED: -126.2.
>
> It is important to note that the training curves of ACCEL and Random show significant variance and fluctuations, while the learning curve of SHED is much smoother. This indicates that SHED can consistently provide suitable environments, enhancing robustness and stability in training.
>
>
> **Q2.** Sampling actions from the teacher's policy during the diffusion process is feasible. Why did you not do this?
>
> **A2.**  First, we provide detailed steps for sampling actions from the teacher's policy during the diffusion process in Appendix B.2 to upsample the teacher's experiences. However, we chose to randomly sample actions to generate synthetic trajectories because it is more straightforward, which is the method we used in our work. Second, using a diffusion model directly as a policy to obtain the teacher's actions would require further modifications to train the diffusion model. This approach is explored in works by Wang et al. (citation number 20) and Janner, Michael, et al. [1]. However, incorporating this into our work would significantly increase its complexity, and therefore, we leave this direction for future work.
> [1] Janner, Michael, et al. "Planning with diffusion for flexible behavior synthesis.". International Conference on Machine Learning. PMLR, 2022.
>
> **Q3.** Could you at least run evaluation on the same levels that ACCEL used? What is the computational demand of running Bipedal Walker for 30K PPO updates? (or at least something like 5k-10k, 50 doesn't seem enough to learn much)
>
> **A3.** As reported in the paper, all models were trained on a single NVIDIA GeForce RTX 3090 GPU and 16 CPUs. Training ACCEL for 30K PPO updates can take approximately 5 days, which is quite long. There seems to be a misunderstanding: we trained on 50 environments, where each environment ran for 4 epochs, and each epoch included 5 PPO minibatches, resulting in a total of 20 PPO updates per environment. Across all 50 environments, there are 1,000 PPO updates. Also, We included some of the levels that ACCEL used, such as SmallCorridor and FourRooms. We will include more levels in the final version.
>
> **Q4.** what details are lacking?
>
> **A4.** Their work is generally well-explained. One difference between their work and ours is that they lack a more detailed technical illustration of how actions and next states are generated using the diffusion model.
>
>
> **Q5.** The SynthER method is a way to upsample data for RL agents, and your teacher is still an RL agent. So I would still argue that it is very similar to what you are doing?
>
> **A5.** SynthER utilizes a diffusion model to generate synthetic experiences from a limited dataset, which is similar to our approach. This work shares similar motivations with ours. However, the key difference is that SynthER focuses on directly synthesizing the experience dataset for the student agent and can be used for online/offline training. In contrast, our approach involves simultaneously upsampling the upper-level teacher's experiences to further assist in training the upper-level teacher agent efficiently, addressing drawbacks in our proposed hierarchical framework. Additionally, directly synthesizing student trajectories to help train the student agents would be unfair in our comparison, as it might induce extra training in terms of the number of PPO updates compared to other approaches. Overall, while we acknowledge that their work and ours share similar motivations and technologies, the application details are different, as we use synthetic trajectories in the upper-level teacher's experience within our hierarchical framework. We will include their work in related work and provide a discussion.

---

> ### Author Response · Authors · 2024-08-10
> **Rebuttal by authors (2)**
>
> **Q6.** Could you run a baseline comparing how well normal ACCEL does that places the blocks directly?
>
> **A6.** Yes, we conducted an experiment using the maze generation method in ACCEL, which involves placing blocks directly to generate the maze. The maze generation process is as follows:
>
> 1. We randomly sample the maze size.
> 2. We randomly place the start and end positions.
> 3. We randomly sample the number of blocks.
> 4. We randomly place the blocks, ensuring that the blocks, start position, and end position are not the same.
> Here are the rough results:
>
> - Randomly sample maze width and height from {8, 9, 10, 11, 12, 13, 14, 15}, randomly sample the number of blocks from [20, 40]:
> Results: ACCEL: -9.8
>
> - Fix maze size at 13*13, randomly sample the number of blocks from [20, 60]:
> Results: ACCEL: -12.7
>
> - Fix the maze size at 13x13, and fix the number of blocks at 60:
> Results: ACCEL: -15.3
>
> For comparison, here are the ACCEL performances under our maze generation method:
> ACCEL: -4.98
>
> Additionally, we added a filter function to exclude mazes without a feasible solution until generate a feasible maze, and the results are:
>
> - Randomly sample maze width and height from {8, 9, 10, 11, 12, 13, 14, 15}, randomly sample the number of blocks from [20, 40]:
> Results: ACCEL: -5.3
>
> - Fix maze size at 13x13, randomly sample the number of blocks from [20, 60]:
> Results: ACCEL: -8.5
>
> - Fix maze size at 13x13, fix the number of blocks at 60:
> Results: ACCEL: -7.8
>
>
> **Q7.** please rectify all of the minor points in your revised manuscript.
>
> **A7.** Thank you for pointing out the minor issues. We will revise the paper to address these points accordingly.

---

> > ### Comment · Reviewer_8F47 · 2024-08-14
> >
> > Thanks for all of your effort in the rebuttal. I have updated my score to 6, on the condition that all of the fixes/explanations are indeed added to the revised paper, and that the paper's claims be slightly softened. I don't think you outperform state of the art UED methods; however, you are competitive with them, and you greatly improve the performance of RL-based methods.

---

### Official Review · Reviewer_kFnL · 2024-07-06

**Soundness:** 2
**Presentation:** 2
**Contribution:** 2
**Rating:** 5
**Confidence:** 3

**Summary:**

The paper presents a novel approach to Unsupervised Environment Design (UED) that addresses the challenges of efficiency by introducing a hierarchical MDP framework and using synthetic data. This framework involves an upper-level RL teacher agent that generates training environments tailored to a lower-level student agent's capabilities. The paper proposes the Synthetically-enhanced Hierarchical Environment Design (SHED) method, which uses generative modeling to create synthetic trajectory datasets, thereby reducing the resource-intensive interactions between agents and environments. The effectiveness of SHED is demonstrated through empirical experiments across various domains, showing superior performance compared to existing UED methods.

**Strengths:**

- The use of diffusion models to generate synthetic trajectories is a novel approach that effectively reduces the computational burden of training the teacher agent.
- The paper provides comprehensive experiments across different domains, demonstrating the effectiveness and robustness of the proposed method compared to state-of-the-art UED approaches.

**Weaknesses:**

- The proposed method introduces significant complexity, particularly in the implementation of the hierarchical MDP and the generative modeling components. This might limit the accessibility and reproducibility of the approach.
- While the empirical results are promising, the evaluation is limited to a few specific domains. It would be beneficial to see broader applicability across more diverse and complex environments.
- Figure 4 is not properly formatted (no values on the axes).

**Questions:**

- Why were diffusion models chosen over other generative models (e.g., GANs, VAEs)? Have other models been considered or tested?
- How well does the proposed method scale to real-world applications with significantly higher complexity and variability? Have any tests been conducted in such settings?

**Limitations:**

The limitations are discussed in Appendix F.1 but I think the authors should discuss the limitations in the main paper.

---

> ### Author Rebuttal · Authors · 2024-08-06
>
> We thank the reviewer for valuable time and feedback, and we kindly request the reviewer to consider our clarifications.
>
> - **W1.** The proposed method introduces significant complexity, particularly in the implementation of the hierarchical MDP and the generative modeling components. This might limit the accessibility and reproducibility of the approach.
>
>   **A1.** We would like to clarify that the two major components introduced in our work, the hierarchical framework and the synthetic trajectories, can be decoupled from each other and can be used independently.  Additionally, the algorithm itself is not overly complex; our hierarchical framework can be viewed as a variant of hierarchical reinforcement learning (HRL),
> The abstractions in SHED are twofold:
> 1. Use performance vector $p(\pi)$ to approximate the embedding of the student policy and also serve as the state for the teacher agent, informing the current student agent’s capability.
> 2. The teacher’s agent is the environment parameters. The teacher’s action $a^u$ is an abstract representation of the next generated environment that is used to train the student agent, allowing the teacher to guide the student’s learning process by setting tailored training environments.
>
> - **W2.** While the empirical results are promising, the evaluation is limited to a few specific domains. It would be beneficial to see broader applicability across more diverse and complex environments.
>
>   **A2.** We acknowledge that this might be perceived as a shortcoming of our study, given our reliance on parameterized environments.  However, we would like to emphasize that our work introduces a novel application of GPT-generated maze environments, a concept that has not been previously explored in UED literature. This innovation enables us to extend traditionally non-parameterized environments to be modeled and controlled by large language models or large multi-modal language models, broadening the potential for future research and applications. Our approach opens up opportunities for further exploration of diverse and complex environments in future studies.
>
> - **W3.** Figure 4 is not properly formatted (no values on the axes).
>
>   **A3.** We have modified the figure. Please see the Figure 1 in the attached PDF document.
>
> - **Q1.** Why were diffusion models chosen over other generative models (e.g., GANs, VAEs)? Have other models been considered or tested?
>
>   **A1.** We chose diffusion models due to their recent success in generating high-quality synthetic data, and we have demonstrated their performance in generating good synthetic trajectories in the Appendix C. While we acknowledge the potential of other generative models, such as GANs and VAEs, we did not test them as our focus was not on comparing the effectiveness of different models for generating synthetic trajectories. Given the strong empirical results achieved by diffusion models, we prioritized exploring their application within our framework.
>
> - **Q2.**  How well does the proposed method scale to real-world applications with significantly higher complexity and variability? Have any tests been conducted in such settings?
>
>   **A2.** Our current study focuses on controlled environments to establish foundational principles, and we plan to test them in real-world applications in future work.
> It should be noted that we are training across environments, so there is a natural stochasticity/complexity that arises in transitions and rewards. That is to say in Minigrid, an agent in one environment may see a different transition/reward than in another environment and it has to still provide a good action that works across both environments.
>
> - **L1.** The limitations are discussed in Appendix F.1 but I think the authors should discuss the limitations in the main paper.
>
>   **A1.** We appreciate your suggestion about discussing the limitations. Initially, we moved this section to the appendix due to page constraints. However, we understand the importance of highlighting these limitations in the main paper, and we will ensure they are included in the revised version.

---

> > ### Comment · Reviewer_kFnL · 2024-08-08
> >
> > Thank you for your detailed response to my review. I will update my score accordingly.

---

### Official Review · Reviewer_EFDP · 2024-07-11

**Soundness:** 2
**Presentation:** 2
**Contribution:** 2
**Rating:** 4
**Confidence:** 3

**Summary:**

This paper considers the Unsupervised Environment Design problem, where a teacher agent seeks to design environments to train a student. Methods such as PLR, PAIRED and ACCEL have recently shown promising performance for random, RL and evolutionary generators. This paper proposes a handful of modifications, using RL with a different objective vs. PAIRED (performance on held out set vs. regret) and also proposes to add synthetic data to accelerate the RL process.

**Strengths:**

* This is an interesting method in a relevant area of research. UED seems to be one of the most active areas of research with plenty of opportunities for impact.
* The use of evaluation environments is sensible and novel.
* The idea of combining this with Genie is incredibly exciting. It would be interesting to hear how this could be possible or could work. Is there any way to show a simple proof of concept?

**Weaknesses:**

* There appear to be two confounding features of the method, the new objective for PAIRED and then the synthetic data. Why do they make sense to combine in this way? It just feels like the authors tried to do "enough for a paper" rather than contribute something meaningful that people can build on. I say this because its unclear how these two independent features interact with other existing algorithms. Maybe we should just do ACCEL with synthetic data for instance? Did the authors try that? If it is in the Appendix already and I missed it then I will increase my score.
* The performance gains are fairly minor, and presented in an unclear fashion with just a bunch of curves on a single plot. Can we get some more rigorous analysis for example using the recommendations from Agarwal et al, "Deep Reinforcement Learning at the Edge of the Statistical Precipice"?
* The Maze experiment seems to have many inductive biases and seems distinct from the diffusion based approach for BipedalWalker and LunarLander. What happens if ACCEL has access to ChatGPT as an editor and then uses replay? This seems like a simpler extension that alone could be a strong paper - although it would resemble ELM (Lehman et al 2022) so it wouldn't be particularly novel.
* The related work is very light. This is disappointing since the paper builds on so many related areas, such as synthetic data, diffusion models, UED, language models for evolution, procedural content generation etc.

**Questions:**

* The authors say "The hierarchical framework can incorporate techniques from existing UED works, such as prioritized level replay". Why did you not do this? Then it would be much clearer to see if it is state of the art.

**Limitations:**

Covered in the Appendix.

---

> ### Author Rebuttal · Authors · 2024-08-06
>
> We thank the reviewer for valuable time and feedback, and we kindly request the reviewer to consider our clarifications.
>
> - **S1.** The idea of combining this with Genie is incredibly exciting.
>
>   **A1.** Genie is the first generative interactive environment trained in an unsupervised manner using unlabelled Internet videos. Genie can generate action-controllable virtual worlds described through text, synthetic images, photographs, and sketches. We propose using a diffusion model to upsample the teacher's experience, which is intuitively compatible, given the diffusion model's proven success in image generation. In our approach, the diffusion model predicts the next student policy representation, similar to how Genie forecasts the next visual frame. Thus, our framework (using a generative model) can be leveraged to learn a simple label that assists Genie in generating more coherent and continuous frames.
>
>
> - **W1.** There appear to be two confounding features of the method, the new objective for PAIRED and then the synthetic data. Why do they make sense to combine in this way? Maybe we should just do ACCEL with synthetic data for instance?
>
>   **A1.** Our primary motivation is to train a student agent with general capabilities more efficiently under resource constraints, specifically with a limited number of environments generated and a limited training horizon compared to the open-ended training in previous UED papers. This is why we employ an RL-based teacher like PAIRED, which generates suitable environments more effectively compared to random generation methods like ACCEL. However, PAIRED uses a regret-based metric—a scalar reward that quantifies the difference between the performances of an expert agent and the current agent. Such metrics can produce difficult environments that are challenging for an RL agent to learn from because regret represents the best-case (upper bound) learning potential, not the actual learning potential of an environment. Consequently, it is hard to fully capture the true general capabilities of student agents. To address this limitation, we propose a hierarchical-MDP framework that better aligns with our objectives and uses the performance vector to approximate the student policy representation.
>
> However, the RL-based teacher approach can be costly in terms of collecting the experience required for training, which conflicts with our motivation to reduce interactions between environments and agents due to limited resources. To mitigate this issue, we propose using a generative model to upsample experiences, thus aiding the training of the RL teacher without incurring significant costly interactions between agents and environments.
>
> While it is theoretically possible to combine ACCEL with synthetic data, this approach would not align with other methods. In all our settings, we ensure that student agents maintain the same level of interaction with environments and have the same number of updates. If we were to use synthetic data with ACCEL, it would create an unfair advantage by providing extra training opportunities for the student agent, and ensuring the same number of interactions would make it difficult to demonstrate the superiority of  using synthetic data with ACCEL, which is why we did not consider combining ACCEL with synthetic data.
>
> We want to highlight that synthetic data can be decoupled from all UED methods, and our hierarchical framework also provides insights into the Hierarchical RL field.
>
>
> - **W2.** Can we get some more rigorous analysis?
>
>   **A2.** Yes. We have provided the aggregate results after min-max normalization (with range=[-21, 1] in the Partially observable Maze domain in Figure 2 in the attached PDF document. Notably, our method SHED dominates all the benchmarks in both the IQM and optimality gap.
>
>
> - **W3.** What happens if ACCEL has access to ChatGPT as an editor and then uses replay?
>
>   **A3.** Thank you for your insightful suggestions. In our Maze experiment, we have already integrated ChatGPT as an editor for ACCEL and utilized replay. Our findings show that ACCEL with ChatGPT and replay demonstrates higher variance in performance, highlighting the instability of random teachers in identifying suitable environments for training student agents.
>
> - **W4.** The related work is very light.
>
>   **A4.** Thank you for highlighting this; we will enhance the related work section to more thoroughly discuss the relevant works in our revised version.
>
> - **Q1.** The authors say "The hierarchical framework can incorporate techniques from existing UED works, such as prioritized level replay". Why did you not do this? Then it would be much clearer to see if it is state of the art.
>
>   **A1.** Our primary motivation is to train a student agent with general capabilities more efficiently under resource constraints, specifically with a limited number of generated environments and a limited training horizon. Integrating prioritized level replay is less appealing in our framework because we aim for our RL teacher to directly generate tailored environments for the current student agent. This approach focuses on dynamic adaptation rather than replay, ensuring that each environment is specifically suited to the student's evolving capabilities.

---

> > ### Comment · Reviewer_EFDP · 2024-08-10
> > **Concerns remain**
> >
> > The issue with this work is what I said in the original review - it is not clear what the motivation is, there are a million different things being compared and no clear take away from the work. The rebuttal hasn't changed anything for me.

---

> ### Author Response · Authors · 2024-08-13
> **Rebuttal by Authors**
>
> Thank the reviewer for the feedback. We understand your concerns and appreciate the opportunity to consider our original motivations.
>
> Our primary motivation, as stated in the abstract and introduction, is to train a student agent with general capabilities more efficiently under resource constraints, specifically with a limited number of generated environments and a limited training horizon. Previous UED papers rely on open-ended training, requiring thousands of generated environments and billions of environment interactions. However, this approach is unrealistic in real-world scenarios due to the limited resources available to construct environments.
>
> To address this challenge, we focus on restricting the number of generated environments (50 environments in our settings) and interaction steps (millions of interactions) to train the student agent to achieve general capabilities. This necessitates designing a teacher agent that can generate suitable environments tailored to the current student's capability level, as opposed to previous methods that relied on random environment generation. Our hierarchical framework addresses this by having the teacher take an approximation of the student policy as the observed state and output environment parameters to generate tailored environments.
>
> However, training the teacher agent is challenging due to the costly collection of the teacher's experience, which requires extensive environment interactions. This conflicts with our original motivation. To mitigate this issue, we propose using a diffusion model to upsample the real collected teacher's experience, reducing the costly interactions needed to gather the teacher's experience.
>
> Additionally, our hierarchical framework can incorporate techniques from existing UED works, such as prioritized level replay. Our proposed module is designed to be decoupled from other techniques. For example, if we have a larger budget for generated environments, we can implement a level buffer to store environments with high learning potential and revisit them in future training.
>
> We hope this explanation clarifies our motivation.

---

### Official Review · Reviewer_AEHi · 2024-07-13

**Soundness:** 2
**Presentation:** 2
**Contribution:** 2
**Rating:** 4
**Confidence:** 3

**Summary:**

The authors of this paper use hierarchical MDP formulation and a teacher agent trained by RL to perform curriculum learning. To address the sparse data available for the teacher agent, this paper uses diffusion models to synthesize datasets for training. This paper performs experiments on lunar lander and bipedal walker environments to validate their claim.

**Strengths:**

Data sparsity is one of the main limitations of using a teacher agent in curricular RL. This paper uses diffusion models to synthesize a dataset for the teacher agent.

**Weaknesses:**

- This paper designs the teacher agent via hierarchical MDP to model the learning process of the student agent to perform curricular RL. However, Fingerprint Policy Optimization (Paul et al, 2019) also has a similar idea of modeling the learning process of the student agent. It would be interesting to explain more about how this paper's idea is related and contributes to this line of thought.

- A fully trained algorithm on the BipedalWalker should approach a cumulative reward of 300. Even the modified version used in the ACCEL paper is measured on a scale of 0 out of 300. However, from Figure 3, it appears all baselines perform less than 50 on the BipedalWalker benchmark. It is questionable whether all baselines were fully trained with the right settings. Also, the performance of the proposed algorithm and those of the baselines are statistically too similar to see whether SHED improves over the baselines in Lunar Lander and BipedalWalker benchmarks. Finally, other than the version of ACCEL in this paper not performing as well as the ACCEL in the original paper, I am curious whether ACCEL can be considered state-of-the-art in the benchmarks as written in line 324. Genetic Curriculum (Song et al, 2022) reports higher cumulative reward on the BipedalWalkerHardcore environment.

- Figure 4 has no scale on timestep and reward.

**Questions:**

- To improve the scores on the review, I hope to see the authors explain how their algorithm works in relation to existing works listed above. I also hope to see clarifications regarding the experiment results stated above as well.

**Limitations:**

The authors has addressed the limitations of this paper.

---

> ### Author Rebuttal · Authors · 2024-08-06
>
> We thank the reviewer for valuable time and feedback, and we kindly request the reviewer to consider our clarifications.
>
> - **Q1.** Fingerprint Policy Optimization (Paul et al, 2019) also models the learning process of the student agent. It would be interesting to explain more about how this paper's idea is related and contributes to this line of thought.
>
>   **A1.** Thank you for pointing out the FPO work, which is very interesting.
>
> FPO utilizes Bayesian optimization to actively select environment variable distributions, thereby enhancing the efficiency of the policy gradient method. It models the environment's effect on policy updates and optimizes for one-step policy improvement by balancing bias and variance. FPO introduces two low-dimensional policy fingerprints:
>
> State Fingerprint: This fingerprint represents the stationary distribution over states induced by the policy. It involves fitting an anisotropic Gaussian to the set of states visited in the trajectories sampled during the estimation of the policy's performance. The size of this fingerprint is equal to the dimensionality of the state space.
> Action Fingerprint: This fingerprint represents the marginal distribution over actions induced by the policy. It is approximated as a Gaussian distribution derived from the actions taken by the policy across various states. The size of this fingerprint corresponds to the dimensionality of the action space.
>
> Both fingerprints serve as low-dimensional proxies for the policy, enabling the Gaussian Process used in Bayesian optimization to efficiently model the relationship between the policy parameters, environment variables, and expected returns. These fingerprints allow FPO to condition its optimization on a compact yet informative representation of the policy, facilitating robust policy learning in environments with significant rare events (SREs).
>
> In contrast, our approach introduces an RL teacher designed to learn the student's learning process and directly generate suitable actions and environments. While FPO uses state fingerprints to represent policy states, we employ a performance vector in the evaluation environments. This approach provides a more accurate representation of the student's general capabilities. Another advantage of using RL is its consideration of long-term rewards, which can significantly enhance the student's general capabilities. Our method aims to optimize the learning trajectory of the student agent by generating suitable environments dynamically to its evolving learning needs.
>
>
> - **Q2.** It is questionable whether all baselines were fully trained with the right settings. Also, the performance of the proposed algorithm and those of the baselines are statistically too similar to see whether SHED improves over the baselines in Lunar Lander and BipedalWalker benchmarks.
>
>    **A2.** The reviewer's point is easy to explain.
> First, our motivation lies in training a student agent with general capabilities more efficiently under resource constraints. Previous algorithms, such as ACCEL, focus on randomly generated environments for open agent training. For example, they train the agent with about 30k PPO updates (1b step), while in our setting, due to the limited generated environment and training horizon, for example, we use only about 1k PPO updates (50 environments) to train the student agent. Therefore, due to resource constraints, there are fewer interactions between the agent and the environment. In this case, the student agent is not the strongest and does not achieve the best performance. Note that we have conducted an ablation study in the appendix, increasing the budget of resource constraints, i.e., more training environments and training horizons. For example, in the left figure of Figure 7, our proposed algorithm SHED can efficiently train the student agent to achieve better general capability in a shorter time horizon, but as the training horizon increases, the performances of ACCEL and SHED converge and tend to be same, which shows that without considering resource constraints, our algorithm can also match the state-of-the-art (SOTA). Under resource constraints, our proposed environment generation process can better generate suitable environments for the current student policy, thereby enhancing its general capability, especially when there is a constraint on the number of generated environments and training horizon.
>
> Second, Figure 3 reflects the average performance over a set of evaluation environments, rather than a single vanilla environment. We consider improving the general capabilities of student agents rather than focusing on a single environment. Please note that our test environments are randomly generated and they are usually much more challenging compared to the vanilla bipedalwalker environment, thus overall performance is much smaller than 300.
>
>
> - **Q3.** Figure 4 has no scale on timestep and reward.
>
>    **A3.** We have modified the figure. Please see the Figure 1 in the attached pdf document.

---

> > ### Comment · Reviewer_AEHi · 2024-08-10
> >
> > Thank you for your response! I believe the Q1 and Q3 are now addressed.
> >
> > However, I'm sorry if I missed the points the authors have already made, but I have some follow up questions regarding the answers to Q2.
> >
> > "30k PPO updates (1b step), while in our setting, due to the limited generated environment and training horizon, for example, we use only about 1k PPO updates (50 environments) to train the student agent."
> > - Would it be OK to rephase the sentence as following? : ACCEL was trained with 1 billion timesteps to interact with the environment and PPO policy was updated 30,000 times. SHED interacted with the environment with 50 environments and had 1,000 PPO updates to the policy.
> >
> > - If the above is true, how many steps did SHED spent interacting with the environment in total for the training?
> >
> > - Why is SHED's interaction measured in number of environments not in steps
> >
> > - When you say 50 environments, are we referring to 50 different obstacle courses in BipedalWalker?
> >
> > - What are the cases that there would be concerns about how many obstacle courses you load for training, instead of how many timesteps the agents spent interacting with the world?
> >
> > - Why does figure 3 report only 1000 training steps when the baselines were trained with a scale of billion steps?

---

> ### Author Response · Authors · 2024-08-10
> **Rebuttal by Authors**
>
> Thank you for your prompt response.
>
> **Q1.** Would it be OK to rephase the sentence as following? : ACCEL was trained with 1 billion timesteps to interact with the environment and PPO policy was updated 30,000 times. SHED interacted with the environment with 50 environments and had 1,000 PPO updates to the policy.
>
> **A1.** Yes, that is true. We will revise the sentence accordingly to improve clarity.
>
> **Q2.** If the above is true, how many steps did SHED spent interacting with the environment in total for the training?
>
> **A2.** SHED interacted with 50 environments. Each environment ran for 4 epochs, and each epoch included 5 PPO minibatches, resulting in a total of 20 PPO updates per environment. Across all 50 environments, this amounted to 1,000 PPO updates. In the LunarLander environment, there were around 1,000,000 total environment steps, while the BipedalWalker environment and Maze environment involved around 10,000,000 and 400,000 total environment steps across all environments, respectively.
>
> **Q3.**  Why is SHED's interaction measured in number of environments not in steps.
>
> **A3.** ACCEL was trained on thousands of environments. We would like to emphasize the motivation of our work is to train a student agent with general capabilities more efficiently under resource constraints, i.e., with a limited number of generated environments. Please note that the number of generated environments is crucial for improving the agent's general capabilities as we only "shallowly" train the agent on each generated environment. Therefore, we use the number of environments as a metric, which provides a clearer understanding of the training conditions. Additionally, we report results in terms of the number of PPO updates to ensure a fair comparison across all approaches with the same number of updates, consistent with other works like PAIRED and ACCEL.
>
> **Q4.** When you say 50 environments, are we referring to 50 different obstacle courses in BipedalWalker?
>
> **A4.** This refers to 50 different instances of the environment that the student agent will be trained in to acquire the general capabilities.
>
>
> **Q5.** What are the cases that there would be concerns about how many obstacle courses you load for training, instead of how many timesteps the agents spent interacting with the world?
>
> **A5.** In general, these two terms are correlated. In this work, we focus on training a student agent with general capabilities more efficiently under resource constraints. The number, diversity, and complexity of the environments are crucial for improving the agent's general capabilities as we only "shallowly" train the agent on each environment. In such cases, the number of environments would be a concern.
>
> **Q6.** Why does figure 3 report only 1000 training steps when the baselines were trained with a scale of a billion steps?
>
> **A6.** There is a misunderstanding, the 1000 training steps refer to the number of PPO updates, not the environment steps. All training algorithms for UED (PAIRED, PLR, ACCEL, and many others) report results in terms of the number of PPO updates. We also do the same in this paper.

---

> > ### Comment · Reviewer_AEHi · 2024-08-12
> >
> > Thank you for your detailed response! I'll update the scores accordingly.

---

> > > ### Author Response · Authors · 2024-08-14
> > > **Rebuttal by authors**
> > >
> > > Thank you for taking the time to read our response. We’re glad that some of your concerns have been addressed. If there are any remaining issues or anything that still needs clarification, please let us know.

---

### Author Rebuttal · Authors · 2024-08-06

We thank the reviewers for their time and valuable feedback! We present the new results in the attached PDF. All feedback will be incorporated into the updated manuscript.

---

### Decision · Program_Chairs · 2024-09-25

**Decision:**

Reject

**Comment:**

This paper introduces a new environment design method.

The reviewers were concerned both about the clarity of the writing, exaggerated claims, and a lack of experimental evaluations.
Some of this was addressed during the rebuttal process, resulting in one reviewer increasing their score. However, other concerns were not addressed, for example the lack of ablations and the fact that the paper makes many changes at once.

While the paper is currently not ready for publication I believe that it will be a good contribution once all of the concerns from the reviewers have been addressed in a future version.